# An experimentally validated network of nine haematopoietic transcription factors reveals mechanisms of cell state stability

Judith Schütte[1,2‡], Huange Wang[1,2], Stella Antoniou[3†], Andrew Jarratt[3†§], Nicola K Wilson[1,2], Joey Riepsaame[3], Fernando J Calero-Nieto[1,2], Victoria Moignard[1,2], Silvia Basilico[1,2], Sarah J Kinston[1,2], Rebecca L Hannah[1,2], Mun Chiang Chan[3], Sylvia T Nürnberg[4,5¶], Willem H Ouwehand[4,5], Nicola Bonzanni[6,7*], Marella FTR de Bruijn[3*], Berthold Göttgens[1,2*]

[1]Department of Haematology, Cambridge Institute for Medical Research, University of Cambridge, Cambridge, United Kingdom; [2]Wellcome Trust - Medical Research Council Cambridge Stem Cell Institute, University of Cambridge, Cambridge, United Kingdom; [3]MRC Molecular Haematology Unit, Weatherall Institute of Molecular Medicine, Radcliffe Department of Medicine, University of Oxford, Oxford, United Kingdom; [4]Department of Haematology, University of Cambridge, Cambridge, United Kingdom; [5]NHS Blood and Transplant, Cambridge, United Kingdom; [6]IBIVU Centre for Integrative Bioinformatics, VU University Amsterdam, Amsterdam, Netherlands; [7]Netherlands Cancer Institute, Amsterdam, Netherlands

*For correspondence: nicola@bonzanni.it (NB); marella.debruijn@imm.ox.ac.uk (MFdB); bg200@cam.ac.uk (BG)

[†]These authors contributed equally to this work

Present address: [‡]Department of Haematology, University Hospital Essen, Essen, Germany; [§]Division of Molecular Medicine, Walter and Eliza Hall Institute of Medical Research, Parkville, Australia; [¶]Perelman School of Medicine, University of Pennsylvania, Philadelphia, United States

Competing interests: The authors declare that no competing interests exist.

**Abstract** Transcription factor (TF) networks determine cell-type identity by establishing and maintaining lineage-specific expression profiles, yet reconstruction of mammalian regulatory network models has been hampered by a lack of comprehensive functional validation of regulatory interactions. Here, we report comprehensive ChIP-Seq, transgenic and reporter gene experimental data that have allowed us to construct an experimentally validated regulatory network model for haematopoietic stem/progenitor cells (HSPCs). Model simulation coupled with subsequent experimental validation using single cell expression profiling revealed potential mechanisms for cell state stabilisation, and also how a leukaemogenic TF fusion protein perturbs key HSPC regulators. The approach presented here should help to improve our understanding of both normal physiological and disease processes.

## Introduction

Tight regulation of gene expression is essential for both the establishment and maintenance of cellular phenotypes within metazoan organisms. The binding of transcription factor proteins (TFs) to specific DNA sequence motifs represents the primary step of decoding genetic information into specific gene expression patterns. TF binding sites (TFBSs) or motifs are usually short (6–10 bp), and therefore found just by chance throughout the genome. Functional TFBSs often occur as evolutionarily conserved clusters, which in the case of enhancers can act over long distances, thus necessitating comprehensive analysis of entire gene loci to understand the transcriptional control mechanisms acting at mammalian gene loci.

Given the complex regulatory circuitries that arise when the control of multiple genes is considered, transcriptional control is often represented in the form of gene regulatory networks (GRNs), which carry most mechanistic information when constructed from detailed knowledge on the TFs

**eLife digest** Blood stem cells and blood progenitor cells replenish a person's entire blood system throughout their life and are crucial for survival. The stem cells have the potential to become any type of blood cell – including white blood cells and red blood cells – while the progenitor cells are slightly more restricted in the types of blood cell they can become. It is important to understand how the balance of cell types is maintained because, in cancers of the blood (also known as leukaemias), this organisation is lost and some cells proliferate abnormally.

Almost all of a person's cells will contain the same genetic information, but different cell types arise when different genes are switched on or off. The genes encoding proteins called transcription factors are particularly important because the proteins can control – either by activating or repressing – many other genes. Importantly, some of these genes will encode other transcription factors, meaning that these proteins essentially work together in networks.

Schütte et al. have now combined extensive biochemical experiments with computational modelling to study some of the transcription factors that define blood stem cells and blood progenitor cells in mice. Firstly, nine transcription factors, which were already known to be important in blood stem cells, were thoroughly studied in mouse cells that could be grown in the laboratory. These experiments provided an overall view of which other genes these transcription factors control. Additional targeted investigations of the nine transcription factors then revealed how these proteins act in combination to activate or repress their respective activities. With this information, Schütte et al. built a computational model, which accurately reproduced how real mouse blood stem and progenitor cells behave when, for example, a transcription factor is deleted. Furthermore, the model could also predict what happens in single cells if the amounts of the transcription factors change.

Lastly, Schütte et al. studied a common type of leukaemia. The model showed that the mutations that occur in this cancer change the finely tuned balance of the nine transcription factors; this may explain why leukaemia cells behave abnormally. In future these models could be extended to more transcription factors and other cell types and cancers.

and the *cis*-regulatory elements with which they interact (*Davidson, 2009*; *Davidson, 2010*; *Petricka and Benfey, 2011*; *Pimanda and Gttgens, 2010*; *Gottgens, 2015*; *Schütte et al., 2012*). Importantly, regulatory network models can provide much more than a representation of existing knowledge, because network simulations can reveal possible molecular mechanisms that underlie highly complex biological processes. Boolean modelling approaches have been used to reconstruct core regulatory networks in blood stem cells (*Bonzanni et al., 2013*) and myeloid progenitors (*Krumsiek et al., 2011*), but neither of these studies took into account the underlying regulatory structure of the relevant gene regulatory elements. Full gene-regulatory information has been used for an ordinary differential equation-based model (*Narula et al., 2010*; *Narula et al., 2013*), but was restricted to a small three-gene core circuit. Large consortia efforts such as ImmGen and FANTOM5 have created comprehensive networks of either regulatory elements or gene signatures important for multipotency and differentiation (*Gazit et al., 2013*; *Jojic et al., 2013*). Furthermore, studies looking at gene regulation circuitry in embryonic stem (ES) cells have proposed regulatory networks important for ES cell identity (*Dunn et al., 2014*; *Zhou et al., 2007*). While the complexities of transcriptional control demand approaches such as network modelling, no single experimental method can provide the complex biological data required for the construction of accurate models. The previously mentioned studies focus their attention on one specific aspect of network modelling and importantly did not combine network analysis with comprehensive functional validation. Given that the key building blocks are gene regulatory sequences and the TFs bound to them, essential information for network reconstruction includes (i) comprehensive TF binding data, (ii) in vivo validation of the functionality of TF-bound regions as bonafide regulatory elements, and (iii) molecular data on the functional consequences of specific TF-binding events (e.g. activation vs. repression). The regulatory network model that we present in this study comprises all of the aforementioned components and is accompanied by functional validation of model predictions.

## Results

### In vivo validation of *cis*-elements as regulatory network nodes connecting 9 HSPC TFs

For the reconstruction of a core GRN model for HSPCs, we focussed on nine major HSPC regulators (ERG, FLI1, GATA2, GFI1B, LYL1, MEIS1, PU.1, RUNX1, TAL1), for which genome-wide binding patterns in the murine multipotent progenitor cell line HPC7 have previously been published (*Wilson et al., 2010*). First, we searched the literature to summarise known *cis*-regulatory regions for the nine TFs that possess haematopoietic activity in transgenic mouse embryos, which recovered a total of 14 regions: *Erg*+85 (*Wilson et al., 2009*), *Fli1*-15 (*Beck et al., 2013*), *Fli1*+12, *Gata2*-3 (*Pimanda et al., 2007*), *Gata2*+3 (= *Gata2*+9.5) (*Wozniak et al., 2007*), *Gfi1b*+13, *Gfi1b*+16, *Gfi1b* +17 (*Wilson et al., 2009*; *Moignard et al., 2013*), *Lyl1* promoter (*Chan et al., 2007*), *Spi1*-14 (*Wilkinson et al., 2014*), *Runx1*+23 (*Nottingham et al., 2007*), *Tal1*-4 (*Gottgens et al., 2004*), *Tal1* +19 (*Göttgens et al., 2002*) and *Tal1*+40 (*Gottgens et al., 2010*).

To extend this partial knowledge of relevant gene regulatory sequences to a comprehensive definition of how these nine TFs might cross-regulate each other, we made use of the genome-wide binding data for the nine TFs (*Wilson et al., 2010*) as well as information on acetylation of histone H3 at lysine 27 (H3K27ac) (*Calero-Nieto et al., 2014*) in the HPC7 blood progenitor cell line. Additional candidate gene regulatory regions for all nine TFs were selected based on the binding of at least three TFs and H3K27ac, since it has been shown previously that transcriptionally active regions are commonly bound by multiple TFs and display H3K27 acetylation (*Hardison and Taylor, 2012*). To assign putative candidate regions to a given TF, they had to be located between its respective upstream and downstream flanking genes, i.e. within the gene body itself or its 5′ and 3′ intergenic flanking regions. The *Erg* gene locus for example contains five candidate *cis*-regulatory regions based on these criteria, namely *Erg*+65, *Erg*+75, *Erg*+85, *Erg*+90 and *Erg*+149 (*Figure 1a*), of which only the *Erg*+85 region had previously been tested in transgenic mice (*Wilson et al., 2009*). Inspection of the gene loci of all nine TFs resulted in the identification of 35 candidate *cis*-regulatory regions (*Figure 1b*, *Figure 1—figure supplements 1–8*). In addition to the 14 haematopoietic enhancers previously published, eight of the 35 new candidate regulatory regions had previously been shown not to possess activity in tissues of the blood system of mouse embryos: *Gata2*-83 (*Gata2*-77), *Gfi1b* promoter (*Moignard et al., 2013*), *Spi1*-18, *Spi1* promoter (*Wilkinson et al., 2014*), *Runx1* P1 promoter (*Bee et al., 2009*), *Tal1*-9, *Tal1* promoter (*Sinclair et al., 1999*) and *Tal1* +6 (*Sánchez et al., 1999*). Of the remaining 27 candidate *cis*-regulatory regions, two coincided with genomic repeat regions (*Runx1*-322 and *Runx1*+1) and were excluded from further analysis because mapping of ChIP-Seq reads to such regions is ambiguous. Since a comprehensive understanding of regulatory interactions among the nine HSPC TFs requires in vivo validation of candidate regulatory regions, we next tested the remaining 25 candidate *cis*-regulatory regions for their ability to mediate reporter gene expression in embryonic sites of definitive haematopoietic cell emergence and colonisation, namely the dorsal aorta and foetal liver of E10.5 to E11.5 transgenic *LacZ*-reporter mouse embryos. For the *Erg* locus, this analysis revealed that in addition to the previously known *Erg*+85 enhancer, the *Erg*+65 and *Erg*+75 regions also displayed activity in the dorsal aorta and/or the foetal liver, while the *Erg*+90 and *Erg*+149 regions did not (*Figure 1c*). Careful inspection of a total of 188 transgenic mouse embryos revealed that nine of the 25 identified regions showed *LacZ* expression in the dorsal aorta and/or foetal liver (*Figure 1b*, *Figure 1—figure supplements 1–8*, *Figure 1—source data 1*). This large-scale transient transgenic screen therefore almost doubled the number of known in vivo validated early haematopoietic regulatory elements for HSPC TFs.

### ChIP-Seq maps for a second progenitor cell line validate core regulatory interactions

Although HPC7 cells are a useful model cell line for the prediction of genomic regions with haematopoietic activity in transgenic mouse assays (*Wilson et al., 2009*), they are refractory to most gene transfer methods and therefore not suitable for functional characterisation of regulatory elements using standard transcriptional assays. By contrast, the 416b myeloid progenitor cell line can be readily transduced by electroporation and therefore represents a candidate cell line for functional dissection of individual regulatory elements. As ChIP-Seq profiles in 416b cells had not been reported

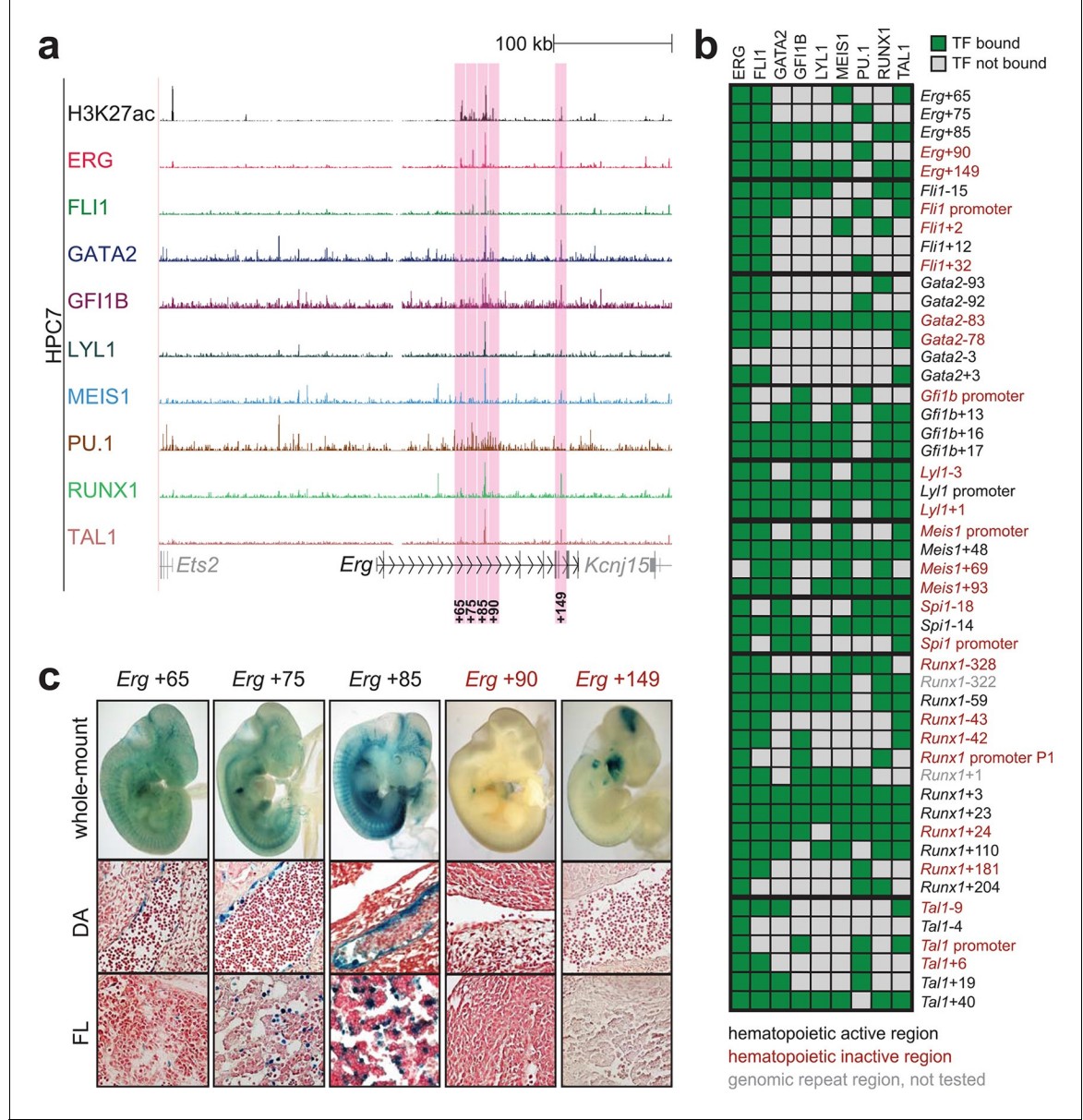

**Figure 1.** Identification of haematopoietic active *cis*-regulatory regions. (a) UCSC screenshot of the *Erg* gene locus for ChIP-Sequencing data for nine haematopoietic TFs (ERG, FLI1, GATA2, GFI1B, LYL1, MEIS1, PU.1, RUNX1 and TAL1 [*Wilson et al., 2010*]) and for H3K27ac (*Calero-Nieto et al., 2014*) in HPC7 cells. Highlighted are all regions of the *Erg* gene locus that are acetylated at H3K27 and are bound by three or more TFs. Numbers indicate the distance (in kb) from the ATG start codon. (b) Summary of the identification of candidate *cis*-regulatory regions for all nine TFs and subsequent analysis in transgenic mouse assays. The inspection of the nine gene loci and the application of the selection criteria (≥3 TFs bound and H3K27ac) identified a total of 49 candidate *cis*-regulatory regions. The heatmap shows the binding pattern of the nine TFs to all candidate regulatory elements in HPC7 cells: green = bound, grey = unbound. Haematopoietic activity in E11.5 transgenic mice is indicated by the font color: black = active, red = not active. Grey indicates genomic repeat regions that were not tested in transgenic mice. Detailed experimental data corresponding to the summary heatmap can be found in *Figure 1* and *Figure 1—figure supplement 1–8*. (c) Haematopoietic activity of the five candidate *Erg* cis-regulatory regions was determined in E11.5 transgenic mouse assays. Shown are X-Gal-stained whole-mount embryos and paraffin sections of the dorsal aorta (DA, ventral side on the left/top) and foetal liver (FL), sites of definitive haematopoiesis. Colour coding as in B.

The following source data and figure supplements are available for figure 1:

**Source data 1.** Number of PCR and LacZ positive transgenic embryos (E10.5–11.5) for each regulatory region.

**Figure supplement 1.** Identification of haematopoietic active *cis*-regulatory elements for *Fli1*.

*Figure 1 continued on next page*

previously, we performed ChIP-Seq for H3K27ac and the nine TFs in this cell line (*Figure 2a*, *Figure 2—figure supplements 1–8*). Alongside with our previously published HPC7 data, this new 416b dataset represents the most complete genome-scale TF-binding analysis in haematopoietic progenitor cell lines to date, with all new data being freely accessible under the following GEO accession number GSE69776 and also at http://codex.stemcells.cam.ac.uk/. Genome-wide TF binding patterns in 416b and HPC7 cells were closely related when compared with binding profiles for the same factors in other haematopoietic lineages (*Figure 2b*, *Figure 2—source data 1*). Inspection of the gene loci for the nine HSPC TFs not only revealed many similarities between 416b and HPC7 cells, but also some differences in TF binding patterns. Overall, TF occupancy at the 23 regions with activity in haematopoietic tissues (14 previously published (*Wilson et al., 2009*; *Beck et al., 2013*; *Pimanda et al., 2007*; *Wozniak et al., 2007*; *Moignard et al., 2013*; *Chan et al., 2007*; *Wilkinson et al., 2014*; *Nottingham et al., 2007*; *Göttgens et al., 2002*; *Gottgens et al., 2004*; *Gottgens et al., 2010*) and 9 newly identified) does not change between the two cell types in 71% of all cases (147 of 207 binding events), is gained in 416b cells in 16% (33 of 207) and lost in 13% (27 of 207) of cases compared to HPC7 cells (*Figure 2c*). Next, all 23 elements were filtered to only retain those elements which were bound by at least 3 of the 9 TFs and displayed elevated H3K27ac in HPC7 and 416 cells. This led to the removal of the *Gata2*-3, which is not bound by any of the nine TFs in either cell type, *Gata2*-92 and *Gfi1b*+13, which are only bound by one or no TFs in 416b cells, and *Fli1*-15, which is not acetylated in 416b cells (*Figure 2c*, *Figure 2—figure supplements 1–3*). Overall, 19 *cis*-regulatory regions were therefore taken forward as a comprehensively validated set of regions for the reconstruction of an HSPC regulatory network model.

## Comprehensive TFBS mutagenesis reveals enhancer-dependent effects of TF binding

The reconstruction of a core regulatory network model requires information about the effect of TF binding on gene expression, which can be activating, repressing or non-functional. In order to analyse the effects of all TF-binding events at all 19 regulatory regions, we performed luciferase reporter assays in stably transfected 416b cells. Based on multiple species alignments between five species (mouse, human, dog, platypus, opossum), we identified conserved TFBSs for the nine TFs (*Figure 3a*, *Figure 3—figure supplements 1–18*, *Figure 3—source data 1*), and generated mutant constructs for each of the 19 regulatory regions, resulting in 87 reporter constructs that were tested by luciferase assays (19 wild-type, 68 mutants). To ensure that DNA binding of the TFs was abrogated, the key DNA bases involved in DNA-protein interactions were mutated and the resulting sequences were scanned to ensure that no new binding sites were created (*Lelieveld et al., 2015*). For each of the 19 regulatory regions, the conserved TFBSs were mutated by family, for example, all six Ets sites within the *Erg*+65 region were mutated simultaneously in one construct, and this element was then treated as the *Erg*+65_Ets mutant. TFBS mutations reduced or increased activity compared to the wild-type enhancer, or indeed had no significant effect (*Figure 3b*, *Figure 3—figure supplements 1–18*). For instance, at the *Erg*+65 region, mutation of the six Ets binding sites or

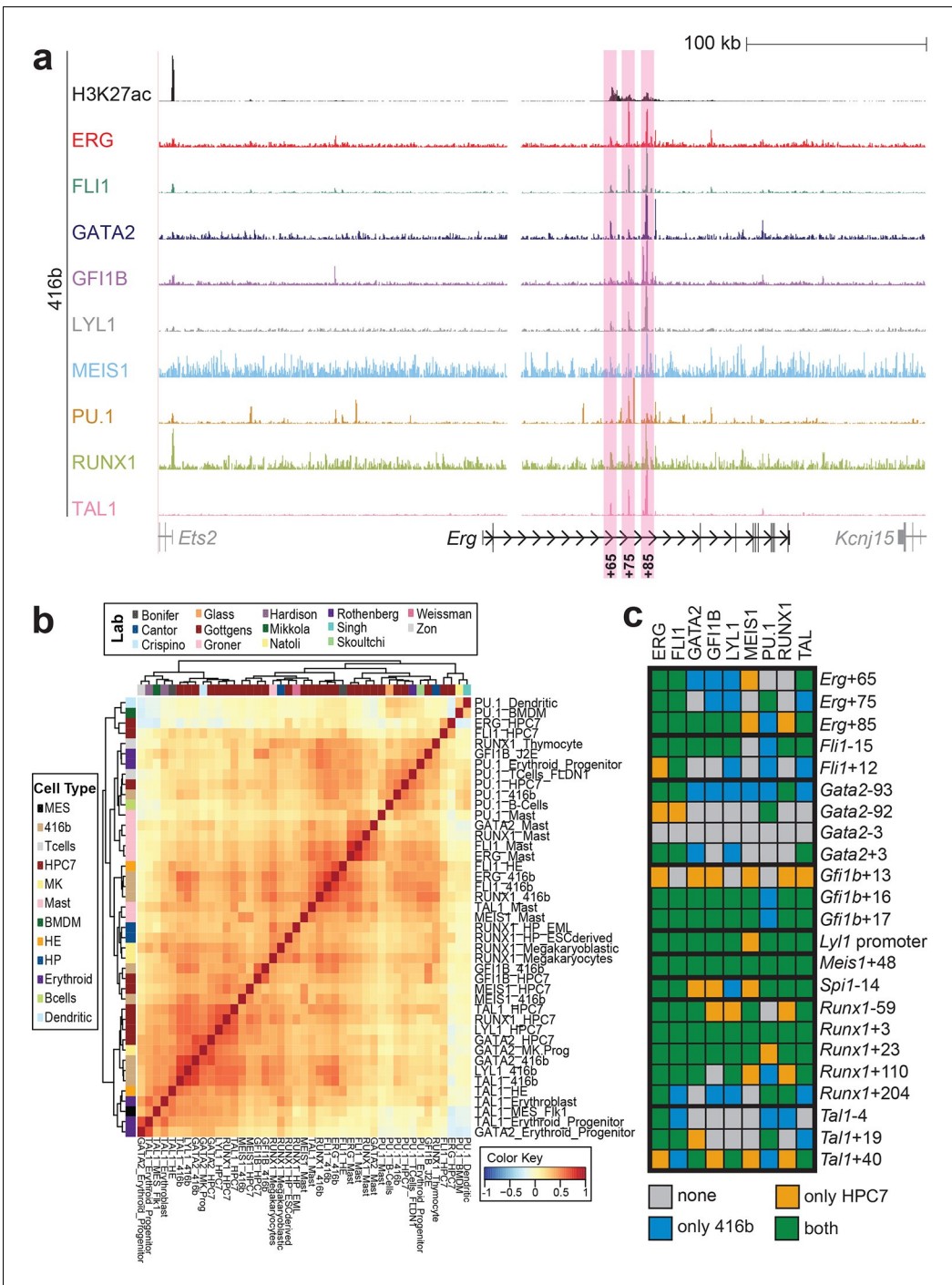

**Figure 2.** Comparison of TF binding pattern at haematopoietic active *cis*-regulatory regions in two haematopoietic progenitor cell lines, HPC7 and 416b. (**a**) UCSC screenshot of the *Erg* gene locus for ChIP-Sequencing data for nine haematopoietic TFs (ERG, FLI1, GATA2, GFI1B, LYL1, MEIS1, PU.1, RUNX1 and TAL1) and for H3K27ac in 416b cells. Highlighted are those haematopoietic active *Erg cis*-regulatory regions that were identified based on acetylation of H3K27 and TF binding in HPC7 cells followed by transgenic mouse assays. Numbers indicate the distance (in kb) from the ATG start codon. (**b**) Hierarchical clustering of the binding profiles for HPC7, 416b and other published datasets. The heatmap shows the pairwise correlation coefficient of peak coverage data between the pairs of samples in the row and column. The order of the samples is identical in columns and rows. Details about samples listed can be found in *Figure 2—source data 1*. (**c**) Pair-wise analysis of binding of the nine TFs to haematopoietic active *cis*-regulatory regions of the nine TFs in HPC7 versus 416b cells.
*Figure 2 continued on next page*

*Figure 2 continued*

Green = bound in both cells types, blue = only bound in 416b cells, orange = only bound in HPC7 cells, grey = not bound in either cell type.

The following source data and figure supplements are available for figure 2:

**Source data 1.** List of ChIP-Seq samples included in the heatmap in *Figure 2b*.

**Figure supplement 1.** UCSC screenshot for the *Fli1* gene locus demonstrating binding patterns for nine key haematopoietic TFs and H3K27ac in 416b cells.

**Figure supplement 2.** UCSC screenshot for the *Gata2* gene locus demonstrating binding patterns for nine key haematopoietic TFs and H3K27ac in 416b cells.

**Figure supplement 3.** UCSC screenshot for the *Gfi1b* gene locus demonstrating binding patterns for nine key haematopoietic TFs and H3K27ac in 416b cells.

**Figure supplement 4.** UCSC screenshot for the *Lyl1* gene locus demonstrating binding patterns for nine key haematopoietic TFs and H3K27ac in 416b cells.

**Figure supplement 5.** UCSC screenshot for the *Meis1* gene locus demonstrating binding patterns for nine key haematopoietic TFs and H3K27ac in 416b cells.

**Figure supplement 6.** UCSC screenshot for the *Runx1* gene locus demonstrating binding patterns for nine key haematopoietic TFs and H3K27ac in 416b cells.

**Figure supplement 7.** UCSC screenshot for the *Spi1* gene locus demonstrating binding patterns for nine key haematopoietic TFs and H3K27ac in 416b cells.

**Figure supplement 8.** UCSC screenshot for the *Tal1* gene locus demonstrating binding patterns for nine key haematopoietic TFs and H3K27ac in 416b cells.

the three Gata binding sites reduced luciferase activity, whereas mutation of the three Ebox or the three Gfi motifs increased luciferase activity (*Figure 3b*). Comparison of the luciferase assay results for all 19 *cis*-regulatory regions (*Figure 3c*) reveals that for each motif class mutation can result in activation, repression or no-change. This observation even extends to single gene loci, where for example mutation of the Gata site reduced activity of the *Erg*+65 region, but increased activity of the *Erg*+85 enhancer (*Figure 3c*). Taken together, this comprehensive mutagenesis screen highlights the dangers associated with extrapolating TF function simply from ChIP-Seq binding events and thus underlines the importance of functional studies for regulatory network reconstruction.

## Dynamic Bayesian network modelling can incorporate complex regulatory information and shows stabilization of the HSPC expression state

We next set out to construct a regulatory network model that incorporates the detailed regulatory information obtained for potential cross-regulation of the nine HSPC TFs obtained in the previous sections (summarised in *Figure 4a*). We focussed on three categories of causal relationships: (i) one or several TFs can bind to a certain type of motif at a given regulatory region, and the probability of a motif being bound depends on the expression levels of the relevant TFs; (ii) TFBS mutations at a given regulatory region altered luciferase activities compared to the wild-type, thus capturing the impact of TF binding on the activity of the given regulatory region; (iii) individual regulatory regions show varying degrees of activation over baseline controls, which translate into different relative strengths of individual *cis*-regulatory regions. To incorporate this multi-layered experimental information, we constructed a three-tier dynamic Bayesian network (DBN) to jointly represent all those causal relationships (see Material and Methods and *Figure 4b*). The reconstructed DBN represents a first-order time-homogeneous Markov process, which is a stochastic process where the transition

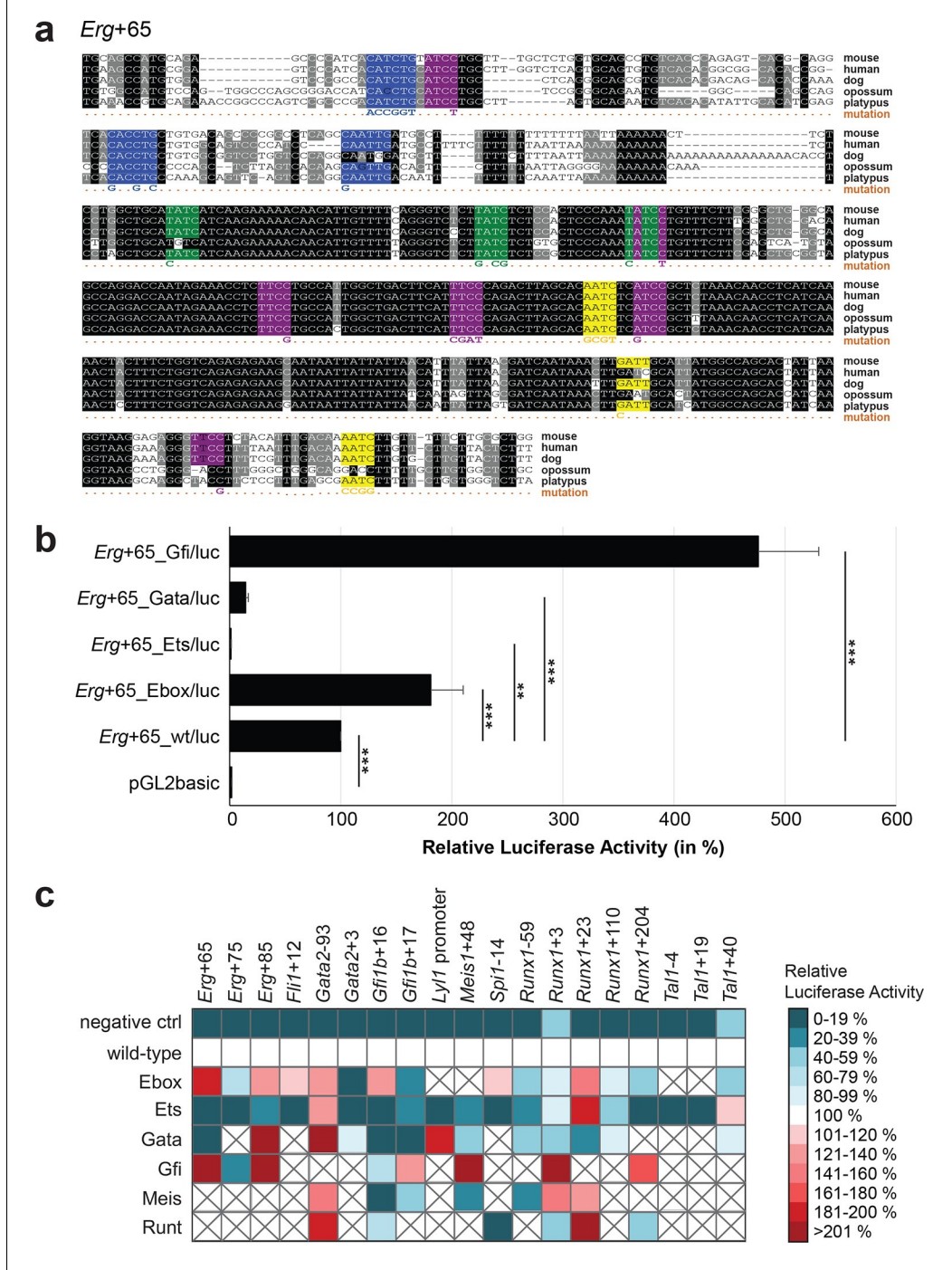

**Figure 3.** TFBS mutagenesis reveals enhancer-dependent effects of TF binding on gene expression. (a) Multiple species alignment of mouse (mm9), human (hg19), dog (canFam2), opossum (monDom5) and platypus (ornAna1) sequences for the *Erg*+65 region. Nucleotides highlighted in black are conserved between all species analysed, nucleotides highlighted in grey are conserved between four of five species. Transcription factor binding sites (TFBS) are highlighted in: blue = Ebox, purple = Ets, green = Gata, yellow = Gfi, red = Meis. The nucleotides that were changed to mutate the TFBSs are indicated below the alignment. All binding sites of one motif family (e.g. all Ebox motifs) were mutated simultaneously. (b) Luciferase assay for the *Erg*+65 wild-type and mutant enhancer in stably transfected 416b cells. Each bar represents the averages of at least three independent experiments with three to four replicates within each experiment. The results are shown relative to the wild-type enhancer activity, which is set to 100%. Error bars represent the standard error of the mean (SEM). Stars indicate significance: **=p-value <0.01, ***=p-value <0.001. p-values were calculated using t-tests, followed by the Fisher's method. (c) Summary of luciferase assay results for all 19 high-confidence haematopoietic active regulatory regions. Relative luciferase activity is illustrated in shades of blue (down-regulation) and red (up-regulation). Crossed-out grey boxes indicate that there is no

*Figure 3 continued on next page*

*Figure 3 continued*

motif for the TF and/or the TF does not bind to the region. Detailed results and corresponding alignments with highlighted TFBSs and their mutations can be found in *Figure 3—figure supplements 1–18*.

The following source data and figure supplements are available for figure 3:

**Source data 1.** List of TF binding sites and the TFs that bind to them.

**Source data 2.** List of co-ordinates and primer sequences for the regulatory regions analysed in this study.

**Figure supplement 1.** Multiple species alignment and luciferase assay results for *Erg*+75.

**Figure supplement 2.** Multiple species alignment and luciferase assay results for *Erg*+85.

**Figure supplement 3.** Multiple species alignment and luciferase assay results for *Fli1*+12.

**Figure supplement 4.** Multiple species alignment and luciferase assay results for *Gata2*-93.

**Figure supplement 5.** Multiple species alignment and luciferase assay results for *Gata2*+3.

**Figure supplement 6.** Multiple species alignment and luciferase assay results for *Gfi1b*+16.

**Figure supplement 7.** Multiple species alignment and luciferase assay results for *Gfi1b*+17.

**Figure supplement 8.** Multiple species alignment and luciferase assay results for *Lyl1* promoter.

**Figure supplement 9.** Multiple species alignment and luciferase assay results for *Meis1*+48.

**Figure supplement 10.** Multiple species alignment and luciferase assay results for *Spi1*-14.

**Figure supplement 11.** Multiple species alignment and luciferase assay results for *Runx1*-59.

**Figure supplement 12.** Multiple species alignment and luciferase assay results for *Runx1*+3.

**Figure supplement 13.** Multiple species alignment and luciferase assay results for *Runx1*+23.

**Figure supplement 14.** Multiple species alignment and luciferase assay results for *Runx1*+110.

**Figure supplement 15.** Multiple species alignment and luciferase assay results for *Runx1*+204.

**Figure supplement 16.** Multiple species alignment and luciferase assay results for *Tal1*-4.

**Figure supplement 17.** Multiple species alignment and luciferase assay results for *Tal1*+19.

**Figure supplement 18.** Multiple species alignment and luciferase assay results for *Tal1*+40.

functions are the same throughout all time points, and the conditional probability distribution of future states depends only on the present state (see Material and Methods). The model is calculated so that the expression at t+1 is influenced by the expression at t0; analogously, the expression at t0 is influenced by the expression at t-1, and so on. Therefore, though the model does not incorporate 'epigenetic memory', past expression levels directly influence current expression levels. Model execution therefore permits the simulation of gene expression states in single cells over time, as well as the calculation of gene expression distributions for each gene across a population of simulated single cells.

Having generated a DBN model incorporating extensive experimental information, we next investigated the expression states following model execution. First, we investigated whether the network

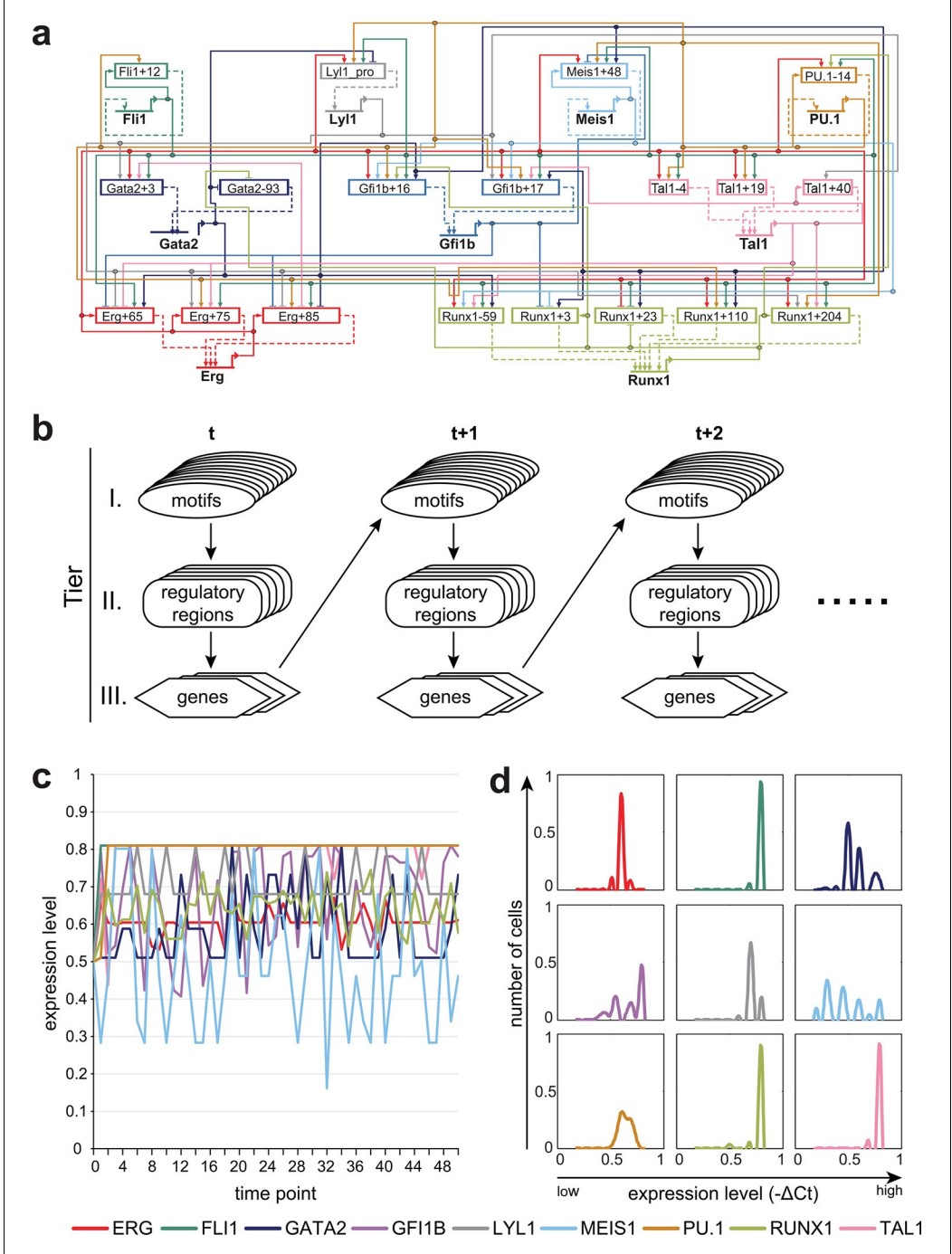

**Figure 4.** A three-tier dynamic Bayesian network (DBN) incorporating transcriptional regulatory information can recapitulate the HSPC expression state. (a) Representation of the complete network diagram generated using the Biotapestry software (*Longabaugh et al., 2005*). (b) Schematic diagram describing the DBN which contains three tiers: I. TF binding motifs within regulatory regions, II. *cis*-regulatory regions influencing the expression levels of the various TFs, and III. genes encoding the TFs. The output of tier III, namely the expression levels of the TF, feed back into the TF binding at the various motifs of tier I. The model therefore is comprised of successive time slices (t). (c) Simulation of a single cell over time. The expression levels of all 9 TFs are the same at the beginning (0.5). The simulation rapidly stabilizes with characteristic TF expression levels. (d) Simulation of a cell population by running the model 1000 times. The scale of the x-axis is linear. Each simulation was run as described in (c).

The following figure supplement is available for figure 4:

**Figure supplement 1.** Simulation of a single cell over time with different expression levels at the beginning.

model was compatible with the HSPC expression profile from which all the experimental data are derived, namely co-expression of all nine TFs. To this end, model execution was initiated with expression levels for all nine TFs set at the midpoint level of 0.5. A representative single cell modelled over time rapidly adopts characteristic levels of expression for each of the nine genes, with some genes showing perpetual fluctuations (*Figure 4c*). The same expression levels were reached when the model was initiated with expression starting at 0.2, 0.8 or with initially only FLI1, RUNX1 and TAL1 being expressed at 0.5 (*Figure 4—figure supplement 1*). We next modelled the overall distribution of the nine TFs as might be seen in a cell population by running 1000 model simulations (*Figure 4d*). This analysis demonstrated that our model is compatible with co-expression of all nine genes within the same single cell. Moreover, stable expression over time for some genes as well as oscillations around a characteristic mean expression level for other genes suggests that our model may have captured those aspects of HSPC regulatory networks that ensure the maintenance of stem/progenitor cells.

## Relative stability to experimental perturbation is recapitulated by the model

The TFs TAL1 and LYL1 are important regulators of adult haematopoiesis, but the deletion of each factor individually has only minor effects on adult HSC function (*Mikkola et al., 2003*; *Hall et al., 2003*; *Capron et al., 2006*). Combined deletion in adult HSCs however causes a severe phenotype with rapid loss of HSPCs (*Souroullas et al., 2009*). We wanted to investigate to what extent our computer model could recapitulate these known phenotypes through in silico perturbation simulations. To quantify if a change in the expression profile of a given TF was significant, we performed a Wilcoxon rank-sum test. Interestingly, this significance calculation demonstrated that both large and small fold-changes can be significant. Simulated perturbation of just LYL1 caused significant alterations to the expression profiles of *Gfi1b, Tal1, Fli1* and *Gata2*, but none of these were associated with a substantial shift in mean expression levels (*Figure 5a*, *Figure 5—figure supplement 1*). Perturbation of just TAL1 caused significant changes to the expression profiles of *Runx1, Gfi1b* and *Gata2*, and again none of these were associated with a substantial shift in expression levels (*Figure 5b*, *Figure 5—figure supplement 1*). Simultaneous deletion of both factors caused significant changes in gene expression profiles in all TFs except for *Fli1*. Unlike for the single TF perturbations, *Gata2* and *Runx1* showed substantial shifts in expression levels when both LYL1 and TAL1 were simulated to be knocked down (*Figure 5c*, *Figure 5—figure supplement 1*). Of note, the significance calculations highlight that there may be no one perfect way to visualize these small fold-change alterations. We therefore also generated histogram plots as an alternative visualization (*Figure 5—figure supplement 2*).

We next wanted to compare model predictions with actual experimental data in the 416b cell line, from which the information for model construction had been derived. Because our DBN model is particularly suited to model the expression states in single cells, we compared predicted and experimentally observed effects of knockdown or overexpression in single cells. To this end, we knocked down the expression of TAL1 in 416b cells by transfecting the cells with siRNA against *Tal1* (siTal1) or control siRNA (siCtrl). Forty-eight hours after transfection, gene expression for the nine network genes was analysed in 44 siTal1 treated cells and 41 siCtrl treated cells. Importantly, 29 of 44 cells (66%) transfected with siTal1 showed no expression of *Tal1* anymore, demonstrating the successful knockdown (*Figure 5d*, *Figure 5—source data 1*). Down-regulation of TAL1 caused a significant change in the expression profiles of *Tal1, Fli1* and *Gfi1b*, but a substantial shift of median expression was only observed for *Tal1* (*Figure 5—figure supplement 1*). Experimental validation therefore confirmed the occurrence of statistically significant small-fold changes in expression profiles following single TF knockdown, although there was no perfect match between the genes affected in the model and experiment. To extend comparisons between model predictions and experimental validation, we investigated the consequences of knocking down the expression of PU.1 and overexpressing GFI1B. Complete removal of PU.1 in silico after the model had reached its initial steady state had no effect on the expression levels of the other TFs (*Figure 6a*). To investigate whether the model prediction is comparable to experimental data obtained from single cells, single cell gene expression analysis using the Fluidigm Biomark HD platform was performed using 416b cells transduced with shRNA against PU.1 (shPU.1) or luciferase (shluc). Three days after transduction, 121 shPU.1 and 123 shluc transduced single cells were analysed for their expression of *Spi1*

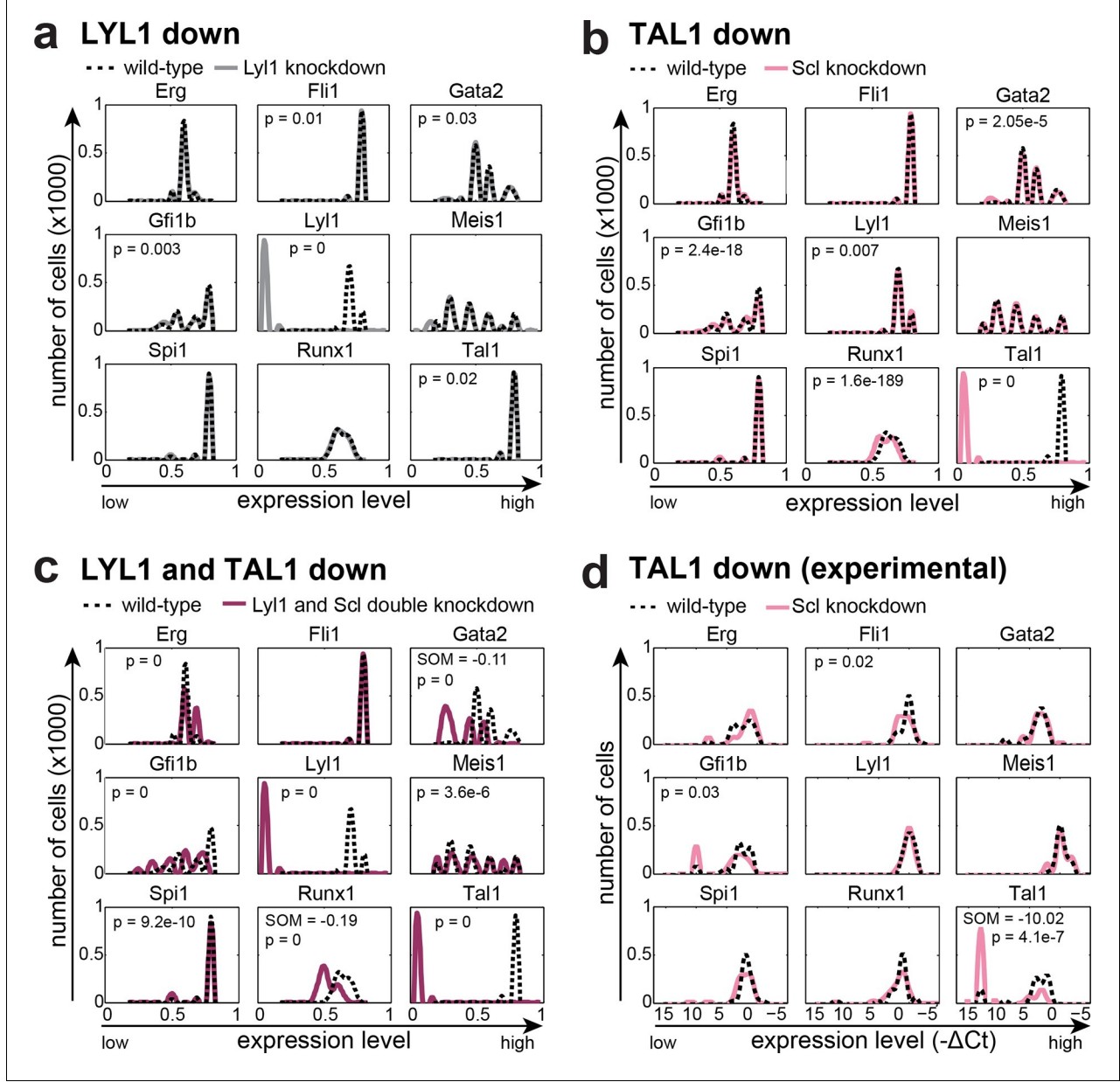

**Figure 5.** The DBN recapitulates the consequences of TAL1 and LYL1 single and double perturbations as seen in vivo and in vitro. Computational prediction of gene expression patterns for the nine TFs of interest after perturbation of TAL1 (**a**), LYL1 (**b**) or both (**c**). Deletion of TAL1 or LYL1 on their own has no major consequences on the expression levels of the other eight TFs of the gene regulatory network, but simultaneous deletion of both TAL1 and LYL1 caused changes in expression of several genes, mainly a decrease in *Gata2* and *Runx1*. This major disruption of the core GRN for blood stem/progenitor cells is therefore consistent with TAL1/LYL1 double knockout HSCs showing a much more severe phenotype than the respective single knock-outs. One thousand simulations were run for each perturbation to determine the TFs expression levels in a 'cell population' by selecting expression levels at random time points after reaching its initial steady state. Expression levels of 0 resemble no expression, whereas expression levels of 1 stand for highest expression level that is possible in this system. The scale of the x-axes is linear. (**d**) Gene expression levels measured in single 416b cells transfected with siRNA constructs against Tal1 or a control. The density plots of gene expression levels after perturbation of TAL1 indicate the relative number of cells (y-axes) at each expression level (x-axes). The scale of the x-axes is linear. The values indicate the results of the Wilcoxon rank-sum test: alterations to the expression profiles are indicated by the p-value (statistical significance: $p < 0.001$ for computational data and $p < 0.05$ for experimental data); substantial shifts in median expression level are indicated by the shift of median (SOM) (SOM $> 0.1$ for computational data and $> 1$ for experimental data). For details, see *Figure 5—figure supplement 1*; for full expression data, see *Figure 5—source data 1* .

The following source data and figure supplements are available for figure 5:

**Source data 1.** Raw and normalised data for the single cell gene expression experiments presented in this study.

*Figure 5 continued*

**Figure supplement 1.** Significance tests for the computational and experimental data after TF perturbations.

**Figure supplement 2.** Histogram plots showing the gene expression distributions of all nine genes of the network for the perturbations presented in this study.

and the other eight TFs of the network. 18 shPU.1-transduced cells (15%) showed a complete loss of *Spi1*, and expression of *Spi1* in the remaining cells was markedly reduced compared to the control cells (shluc) (*Figure 6a*, *Figure 5—source data 1*), highlighting the efficiency of the PU.1 knock-down. *Spi1, Runx1, Erg* and *Fli1* showed a significant change in expression profiles after the depletion of PU.1, but this involved a substantial shift in median expression levels only for *Spi1* and *Runx1* (*Figure 5—figure supplement 1*). Expression profiles of the remaining five TFs did not change as a result of reduced PU.1 levels (*Figure 6a*, *Figure 5—source data 1*), therefore mostly confirming the model prediction.

Next, we modelled GFI1B overexpression in silico by increasing the expression level of *Gfi1b* to the maximum value after the model had reached its initial steady state which led to a significant change in the expression profiles of *Gfi1b, Meis1, Erg* and *Runx1*, although only *Gfi1b* and *Meis1* showed a substantial shift in median expression levels (*Figure 6b*, *Figure 5—figure supplement 1*, *Figure 5—source data 1*). Expression profiles of the other five TFs were unaltered. Single cell gene expression analysis of 90 single 416b cells transduced with a GFI1B-expressing vector and 104 single 416b cells transduced with an empty control vector showed a significant increase in the expression of *Gfi1b* and a significant alteration to the expression profile of *Erg*, but only the changes to *Gfi1b* involved a substantial shift in median expression levels. No significant expression changes were seen in any of the other seven network genes (*Figure 6b*). Both PU.1 and GFI1B perturbation studies therefore emphasize the resilience of the HSPC TF network to single TF perturbation. Moreover, our in silico model reflects this, thus suggesting that the comprehensive experimental information used to construct the network model has allowed us to capture key mechanistic aspects of HSPC regulation. Of note, there were no short-term major expression changes immediately after the perturbation in the in silico simulations for the three single TF perturbations described above. For completeness, we performed in silico modelling for all permutations of single TF knockdown / overexpression as well as all pairwise combinations of all 9 TFs analysed (a total of 162 simulations, *Figure 6—source data 1*).

## Major perturbations by the AML-ETO oncoprotein are captured by the network model

As the TF network described above is relatively stable to single TF perturbations, we set out to test whether a simulation that mimics the situation present in leukaemic cells can influence the expression states of the nine TFs in our network. The *Aml-Eto9a* translocation is amongst the most frequent mutations in AML (reviewed in [*Licht, 2001*]). The resulting fusion protein is thought to act in a dominant-negative manner to repress RUNX1 target genes. To simulate the leukaemic scenario caused by AML-ETO expression, we fixed the level of *Runx1* to be the maximum value 1 and at the same time converted all activating inputs of RUNX1 to inhibiting inputs in our DBN model. Interestingly, this simulation of a 'leukaemic' perturbation caused significant expression changes to all eight of the core HSPC TFs (*Figure 6c*). To compare the AML-ETO simulation results with experimental data, we utilised a doxycycline-inducible expression system to generate 416b cells with inducible expression of AML1-ETO fused to a mCherry reporter via a self-cleaving 2A peptide spacer. Following doxycycline induction, 56 single mCherry positive and 122 single mCherry negative 416b cells were analysed by single cell gene expression. Significant gene expression changes can be seen in six of the nine core HSPC TFs (all except *Tal1, Erg* and *Gata2*) thus highlighting significant overlap between predictions and experimental validation, although there are also notable differences between model predictions and the experimental data (see for example Gata2; *Figure 6c*, *Figure 5—figure supplement 1*, *Figure 5—source data 1*). These results demonstrate that our new HSPC network model can capture many gene expression changes caused by ectopic expression of a leukaemia oncogene

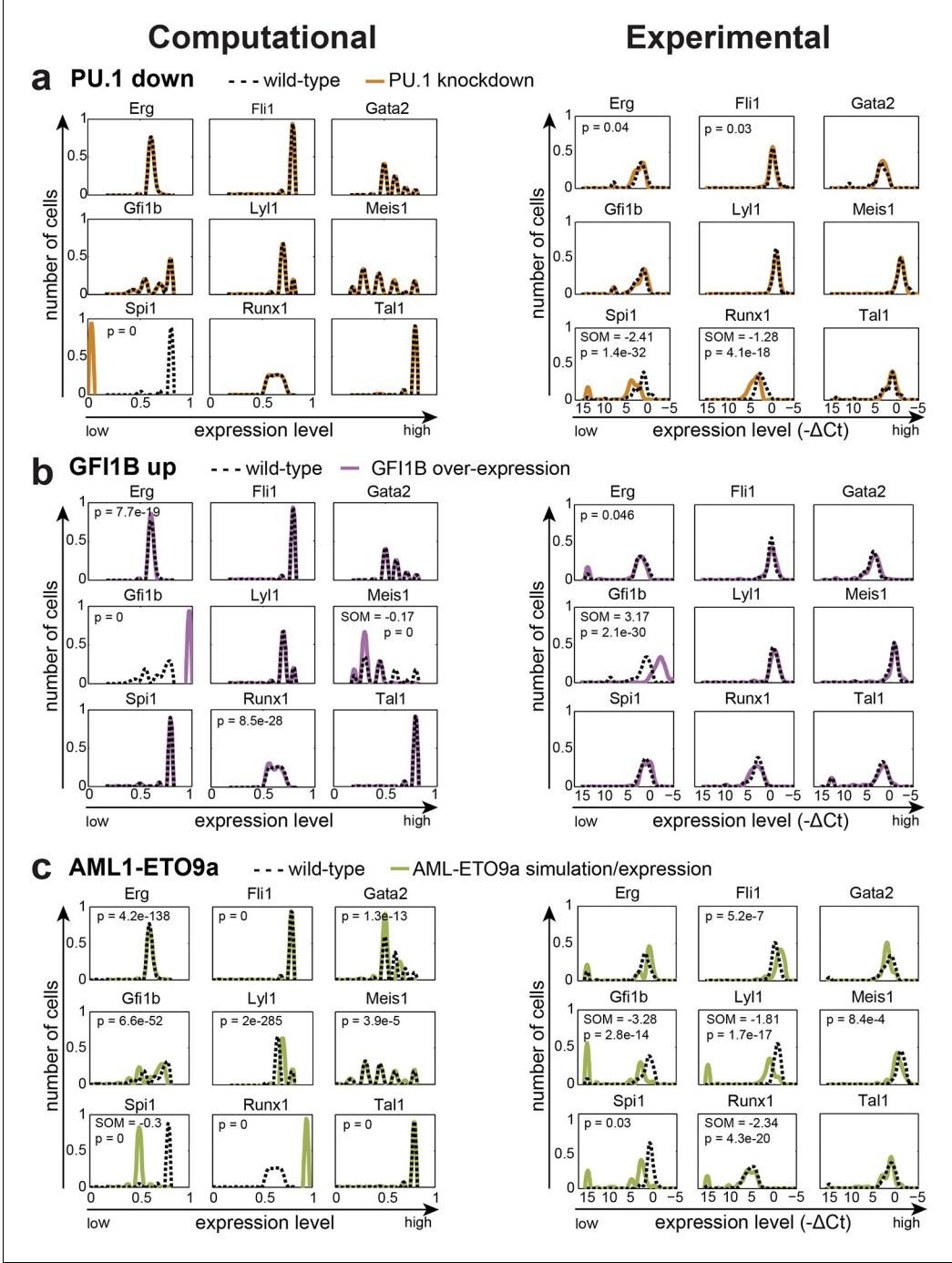

**Figure 6.** The DBN captures the transcriptional consequences of network perturbations. Left panel: Computational prediction of gene expression after perturbation of specific TFs. 1000 simulations were run for each perturbation to determine expression levels in a 'cell population' (expression at 0 resembles no expression, whereas expression of 1 represents the highest possible expression level). The scale of the x-axes is linear. Right panel: Density plots of gene expression levels in single 416b cells after perturbation of specific TFs indicating the relative number of cells at each expression level. The scale of the x-axes is linear. The values indicate the results of the Wilcoxon rank-sum test: alterations to the expression profiles are indicated by the p-value (statistical significance: $p<0.001$ for computational data and $p<0.05$ for experimental data); substantial shifts in median expression level are indicated by the shift of median (SOM) (SOM $>0.1$ for computational data and $>1$ for experimental data). For details, see *Figure 5—figure supplement 1*. (*a*) PU.1 down-regulation: (Left) Computational prediction of gene expression after PU.1 knockdown (Spi1 was set to 0 after reaching its initial

*Figure 6 continued on next page*

*Figure 6 continued*

steady state). (Right) Gene expression levels measured in single 416b cells transduced with shRNA constructs against shluc (wild-type) or shPU.1 (PU.1 knockdown). (b) GFI1B overexpression: (Left) Computational prediction of gene expression after overexpression of GFI1B (Gfi1b was set to 1 after reaching its initial steady state). (Right) Gene expression levels in single 416b cells transduced with a GFI1B-expressing vector compared to an empty vector control (wild-type). (c) Consequences of the AML-ETO9a oncogene: (Left) Computational prediction of gene expression patterns after introducing the dominant-negative effect of the AML-ETO9a oncogene (Runx1 was fixed at the maximum value of 1 after reaching its initial steady state and in addition all Runt binding sites were set to have a repressive effect). (Right) Gene expression levels measured in single 416b cells transduced with an AML-ETO9a expressing vector fused to mCherry. mCherry positive cells were compared to mCherry negative cells (wild-type).

The following source data is available for figure 6:

**Source data 1.** Summary of all computational simulations for perturbations of one or two TFs.

as well as providing a useful model for normal HSPC transcriptional regulation. The inability of any model to completely recapitulate experimental data is not unexpected. Possible reasons in our case may include more complex activities of the onco-fusion protein than would be captured by our assumption that its 'only' function is as a straightforward dominant-negative effect, or the fact that the computational model is a closed system of only the 9 network TFs, whereas the experimental single cell perturbation is subject to possible knock-on consequences from gene changes outside of the 9 TF network.

## Discussion

Transcription factor networks are widely recognised as key determinants of cell type identity. Since the functionality of such regulatory networks is ultimately encoded in the genome, the logic that governs interactions between network components should be identifiable, and in due course allow for the construction of network models that are capable of capturing the behaviour of complex biological processes. However, the construction of such network models has so far been severely restricted because the identification and subsequent functional characterisation of mammalian regulatory sequences represent major challenges, and the connectivity and interaction rules within regulatory networks can be highly complex. Here, we report a comprehensive mammalian transcriptional network model that is fully grounded in experimental data. Model simulation coupled with subsequent experimental validation using sophisticated single cell transcriptional assays revealed the mechanistic basis for cell state stability within a haematopoietic progenitor model cell line, and also how a leukaemogenic TF fusion protein can perturb the expression of a subset of key blood stem cell regulators.

Pictorial representations of putative network models are commonly shown in publications reporting ChIP-Seq TF binding datasets (*Tijssen et al., 2011*). However, due to the lack of experimental underpinning, such representations are simple images that do not encode any of the underlying gene regulatory logic, and importantly therefore cannot provide executable computational models that can be used to simulate biological systems. Although the experimentally-grounded network model shows good agreement with the relative expression states of the nine TFs for the wild-type as well as the perturbation data, model predictions are not correct in all cases. Apart from the obvious caveat that any computer model is an abstraction of reality and therefore will not be correct in every detail, it also needs to be stressed that we treat the nine TFs as an isolated module for the computer simulations, and therefore could not account for possible influences by additional genes that may affect single cell gene expression measurements in the perturbation experiments.

Statistical significance calculations demonstrated that both the computer model and the experimental data showed significant changes in gene expression profiles that were associated with minimal fold-change alterations to median expression levels. Such alterations to expression profiles were prevalent in both single and double gene perturbations, whereas substantial shifts in median expression were mostly restricted to the double perturbations (and also the AML-ETO oncogene overexpression). This observation suggests that (i) our approach has the capacity to reveal the aspects of

the fine-grained nature of biological networks, and (ii) the network presented in this study is largely resistant to perturbations of individual TFs in terms of substantial fold-change alterations in median expression levels. We believe that it may well be possible that the statistically significant small-fold changes in HSPC network genes may be responsible for the mild phenotypes seen when major HSPC regulators are deleted in adult HSPCs. $Tal1^{-/-}$ mice for example are not viable because TAL1 is absolutely required for embryonic blood development (*Shivdasani et al., 1995*), yet deletion of TAL1 in adult HSCs only causes minor phenotypes (*Mikkola et al., 2003*). Another noteworthy observation is that it would have been impossible to detect the statistically significant yet small fold-changes using conventional expression profiling, because they only become apparent following the statistical analysis of expression distributions generated by assaying lots of single cells. More generally, it is important to acknowledge that the question of how close the present model comes to capturing the underlying biological processes can only be revealed through much more exhaustive experimental validation studies.

A potential caveat for network reconstruction based on identification of regulatory elements comes from the difficulties associated with capturing negative regulatory elements. As shown elegantly for CD4 and CD8 gene silencers in the lymphoid lineage, TFs involved in the early repression of a locus are not required for the maintenance of the silenced state (*Taniuchi et al., 2002a*; *Taniuchi et al., 2002b*). Identification of negative regulatory inputs may therefore require an expansion of datasets to look across sequential developmental stages. It will therefore be important in the future to extend the work presented here to include additional HSPC regulators as well as additional stages along the haematopoietic differentiation hierarchy. Of note, TF-mediated cellular programming experiments have demonstrated that the modules of 3–4 TFs are able to confer cell-type specific transcriptional programmes (*Takahashi and Yamanaka, 2006*; *Graf and Enver, 2009*; *Batta et al., 2014*; *Riddell et al., 2014*), consistent with the notion that a network composed of nine key HSPC regulators is able to capture useful information about HSPC regulatory programmes.

One of the most striking observations of the regulatory network defined here is the high degree to which the HSPC expression state is stabilised. As such, this model is different from previous experimentally-grounded transcriptional regulatory network models (*Peter and Davidson, 2011*). These earlier model organism networks have inherent forward momentum, where the model captures the progression through successive embryonic developmental stages characterised by distinct expression states.

The model reported here is based on and validated with data from haematopoietic progenitor cell lines, which can differentiate (*Pinto do O et al., 1998*; *Dexter et al., 1979*), but can also be maintained in stable self-renewing conditions. A recent study by Busch and colleagues tracked labelled Tie2$^+$ HSCs in the bone marrow, and showed that haematopoietic progenitors in vivo are also characterised by a substantial self-renewal capability, therefore highlighting the stable state in which they can reside for several months (*Busch et al., 2015*). The observed stability of the HSPC expression state presented here is therefore likely to capture aspects of the regulatory mechanisms maintaining the steady state of primary haematopoietic progenitor cells, a notion reinforced further by the fact that our model is based on in vivo validated regulatory elements.

The two types of models therefore accurately capture the properties of the distinct biological processes, e.g. driving developmental progression on the one hand, and maintaining a given cellular state on the other. Different design principles are likely to be at play, with feed-forward loops representing key building blocks of early developmental GRNs, while the network described here shows an abundance of auto-regulatory feedback loops and partially redundant enhancer elements, both of which may serve to stabilise a given cellular state.

Of particular interest may be the organisation of the *Runx1* gene locus, where RUNX1 protein provides positive feedback at some, and negative feedback at other HSPC enhancers. Given that these different enhancers employ overlapping yet distinct sets of upstream regulators, it is tempting to speculate that such an arrangement not only stabilises a given expression level, but also provides the means to either up- or down-regulate RUNX1 expression in response to diverse external stimuli that may act on specific RUNX1 co-factors at either the repressing or activating RUNX1 binding events.

Taken together, we report widely applicable experimental and computational strategies for generating fully validated regulatory network models in complex mammalian systems. We furthermore demonstrate how such a model derived for blood stem/progenitor cells reveals mechanisms for

stabilisation of the progenitor cell state, and can be utilised to analyse core network perturbations caused by leukaemic oncogenes.

## Materials and methods

### ChIP-Sequencing and data processing

The mouse myeloid progenitor 416b cell line (*Dexter et al., 1979*) was received from Chester Beatty lab and confirmed to be mycoplasma free. The cells were cultured in RPMI with 10% FCS and 1% Penicillin/Streptomycin.

ChIP assays were performed as previously described (*Wilson et al., 2009*; *Calero-Nieto et al., 2014*), amplified using the Illumina TruSeq ChIP Sample Prep Kit and sequenced using the Illumina HiSeq 2500 System following the manufacturer's instructions. Sequencing reads were mapped to the mm10 mouse reference genome using Bowtie2 (*Langmead and Salzberg, 2012*), converted to a density plot and displayed as UCSC genome browser custom tracks. Peaks were called using MACS2 software (*Zhang et al., 2008*). Mapped reads were converted to density plots and displayed as UCSC genome browser custom tracks. The raw and processed ChIP-Seq data have been submitted to the NCBI Gene Expression Omnibus (www.ncbi.nlm.nih.gov/geo) and assigned the identifier GSE69776. A binary binding matrix was created using in-house scripts, clustered using the dice coefficient and a heatmap was plotted using gplots in R to compare newly generated ChIP-Seq data with previously published data (*Sanchez-Castillo et al., 2015*).

### Analysis of enhancer activity in transient transgenic mouse embryos

Genomic fragments spanning the candidate *cis*-regulatory regions were generated by PCR or ordered as gBlocks (Life Technologies GmbH, Germany) and cloned downstream of the LacZ gene in an hsp68LacZ (Runx1 constructs) or SVLacZ (all other constructs) reporter vector. Coordinates of candidate chromosomal regions and corresponding primer sequences are given in *Figure 3—source data 2*. For Runx1, E10 mouse transient transgenic embryos carrying LacZ enhancer-reporter constructs were generated by pronuclear injection of (C57BL/6 x CBA)/F2 zygotes following standard procedures. Transgenic embryos were identified by LacZ-specific PCR on genomic DNA isolated from yolk sac (5'-GCAGATGCACGGTTACGATG-3'; 5'-GTGGCAACATGGAAATCGCTG-3'). Xgal staining and cryostat sectioning were performed as previously described (*Nottingham et al., 2007*). Embryos were photographed using a Leica MZFLIII microscope, Leica DFC 300F digital camera (Leica Microsystems, Milton Keynes, UK) and Openlab software (Improvision, Coventry, UK) and sections were examined using a Nikon Eclipse E600 microscope (Nikon, Japan) equipped with 20x and 40x Nomarski objectives. Photographs were taken using a Nikon DXM 1200c Digital Camera (Nikon, Tokyo, Japan). E11.5 transient transgenic embryos of all other candidate *cis*-regulatory regions were generated by Cyagen Biosciences Inc (Guangzhou, China). Whole-mount embryos were stained with 5-bromo-4-chloro-3-indolyl-β-d-galactopyranoside (X-Gal) for β-galactosidase expression and photographed using a Nikon Digital Sight DS-FL1 camera attached to a Nikon SM7800 microscope (Nikon, Kingston-upon-Thames, UK). Candidate transgenic mouse embryos with LacZ staining in haematopoietic tissues were subsequently embedded in paraffin, stained with 0.1% (w/v) Neutral Red and cut into 6 µm deep longitudinal sections. Images of sections were acquired with a Pixera Penguin 600CL camera attached to an Olympus BX51 microscope. All images were processed using Adobe Photoshop (Adobe systems Europe, Uxbridge, United Kingdom).

### Luciferase reporter assays

Wild-type and mutant DNA fragments for candidate regulatory regions were either cloned using standard recombinant DNA techniques, ordered as gBlocks (Life Technologies) or obtained from GeneArt by Life Technolgies. DNA fragments were cloned into pGL2 basic or pGL2 promoter vectors from Promega using restriction enzymes or by Gibson Assembly. TFBSs for the nine TFs of interest (corresponding DNA sequences are listed in *Figure 3—source data 1*) were identified based on multiple species alignments between five species (mouse, human, dog, platypus, opossum). Where a region contained multiple instances of the same motif, a single mutant construct with all relevant motifs mutated simultaneously was generated (for generated point mutations check *Figure 3a* and *Figure 3—figure supplements 1–18*). Where TF binding was observed in ChIP-Seq experiments in

416b cells, but the TFBS was not conserved, the motifs present in the mouse sequence were mutated. Stable transfections of the 416b cell line were performed using 10 µg reporter construct, 2 µg neomycin resistance plasmid and $1x10^7$ 416b cells in 180 µl culture medium per pulse. The sample was electroporated at 220 V and capacitance of 900 µF using the GenePulser Xcell Electroporation System (Bio-Rad, United Kingdom). Immediately after transfection, the sample was split into four culture plates. Twenty-four hours after transfection Geneticin G418 (Gibco by Life Technologies) at a final concentration of 0.75 mg/ml was applied to the culture to select for transfected cells. The activity of the luciferase reporter constructs was measured 12–16 days after transfection by using a FLUOstar OPTIMA luminometer (BMG LABTECH, United Kingdom). The luciferase activity was normalised to the cell number and presented as relative activity compared to the wild-type construct. All assays were performed at least three times in quadruplicates.

## Single cell gene expression and data analysis

The TAL1 knockdown was performed using pools of siRNA against Tal1 (Dharmacon, United Kingdom) which were transfected into 416b cells. Briefly, $1 \times 10^6$ cells were electroporated with either a control or Tal1 siRNA. Forty-eight hours after transfection, cells were sorted into 96-well PCR plates containing lysis buffer using the BD Influx Cell Sorter.

The PU.1 knockdown was performed as previously described (*Calero-Nieto et al., 2014*).

The MigR1-Gfi1b retroviral expression vector and the corresponding empty vector control (*Xu and Kee, 2007*) were used for GFI1B overexpression. Two million 416b cells were transduced with the above listed vectors by adding viral supernatant and 4 µg/ml polybrene to the cells, followed by centrifugation at 900 x g for 90 min at 32°C and incubation with 5% $CO_2$ at 32°C. Half of the media was then replaced with fresh culture media, and cells were incubated at 37°C with 5% $CO_2$. Forty-eight hours after transduction, GFP$^+$ cells for each cell population were sorted into 96-well PCR plates containing lysis buffer using the BD Influx Cell Sorter.

To induce AML1-ETO9a expression, the 416b cell line was co-transfected with: 1) a plasmid containing the tetracycline transcription silencer (tTS), the tetracycline transactivator (rtTA) and blasticidine resistance under the control of the *EF1α* promoter; 2) a plasmid containing the entire *Aml-Eto9a* cDNA (obtained from vector MigR1-AE9a, Addgene no. 12433) in frame with a F2A element and the mCherry protein under the control of a tetracycline responsive element; and 3) transposase PL623 (*Wang et al., 2011*) (kindly donated by Pentao Liu, Sanger Institute, Cambridge) to promote simultaneous stable integration of the two constructs described above. After 6 days of culture without selection, cells were incubated with 1 µg/ml of Doxycycline for 24 hr and then stained with DAPI. mCherry positive and negative cells that did not stain with DAPI were sorted into 96-well PCR plates containing lysis buffer using the BD Influx Cell Sorter.

Single cell gene expression analysis was performed using the Fluidigm BioMark platform followed by bioinformatics analysis as previously described (*Moignard et al., 2013*). All cells that express less than 48% of genes assayed were removed from the analysis for PU.1 knockdown and GFI1B overexpression, all cells expressing less than 56% of genes assayed were removed from the TAL1 knockdown and all cells that express less than 44% of genes assayed were removed from the analysis for the AML-ETO9a induction. Importantly, this thresholding resulted in the removal of similar numbers of cells in both the perturbation and control arms of the experiments. The raw data as well as the normalised data (normalised to Ubc and Polr2a) of the gene expression analysis are listed in *Figure 5—source data 1*).

## Computational modelling

The first-order DBN shown in *Figure 4b* was established on the basis of regulatory information summarized in *Figure 4a*. The DBN essentially presents a discrete-time stochastic process that has the Markov property, i.e. the state of the process at the next time point depends purely on its state at the current time point. Also note that this is a time-homogeneous (or time-invariant) DBN, where the transition functions/matrices are the same throughout all time points.

To specify parameters of the DBN, we defined a motif family at a specific regulatory region as a unique binary variable; with value '1' indicating that no motif of a motif family is bound at the specific region and value '2' indicating that at least one motif of the motif family is bound by a TF at this region. We assumed that any of the following three factors can lead to a higher probability of a

motif being bound by a TF and therefore taking the value 2: (i) more motifs of the same type present within a regulatory region; (ii) multiple TFs that can bind to the same motif, such as TAL1 and LYL1 both binding to Eboxes; (iii) higher expression levels of the TFs. The probabilities were thus calculated based on these three sources of information (see below for an example). We next defined that every regulatory region was a continuous variable on the close interval [0, 1], and its value was determined by the accumulated effects of all motifs present within the regulatory region. Finally, the expression levels of the nine TFs were also defined as continuous variables ranging from 0 to 0.8, and their expression levels were determined by the accumulated activities of the relevant regulatory regions.

Considering that variables in the top tier of the DBN are binary whereas those in the middle and bottom tiers are continuous, we found conditional linear Gaussian distribution (*Koller and Friedman, 2009*) to be an appropriate generic representation of the intra-slice conditional probability distributions. Specifically, the regression coefficient of a regulatory region on a motif family was estimated by normalizing the logarithmic deviation of luciferase activity, where deviation refers to the change of luciferase activity between the wild-type and the mutated (one motif family at a time) regulatory region (see below for a demonstration). Using the logarithmic deviation allowed us to account for the differences in effect sizes of various motif families by rescaling the differences to a comparable range. Similarly, for each of the nine genes, the regression coefficient of its expression level on a relevant regulatory region was estimated by normalizing the logarithmic deviation of luciferase activity, where deviation refers to the change of luciferase activity compared to the empty vector controls. All Matlab source codes are available at https://github.com/Huange and also at http://burrn-sim.stemcells.cam.ac.uk/.

## Detailed explanation of the modelling of each tier of the DBN
### a) Estimating the discrete probability distribution of a motif variable

The probability of a motif family at a given regulatory region taking value 1 or 2 (i.e. being unbound or bound) was calculated based on: (i) the number of such motifs in that regulatory region; (ii) the expression levels of the relevant TFs.

For example, three Ebox motifs were found at *Erg*+65 (*Figure 3a*). They can be bound by either TAL1 or LYL1. Thus, we assigned that $P$(Ebox@Erg+65=1) and $P$(Ebox@Erg+65=2) were determined by {3, TAL1, LYL1}. We assumed that (i) the expression level of a TF is proportional to the probability of that TF binding to a target motif; and (ii) the bindings of TFs to multiple motifs are independent events. Gene expression levels were defined within the closed interval [0, 1], which is identical to the possible range of probabilities. For ease of calculation, we took the expression level of a TF as its probability of binding to a motif. Accordingly, we have

$$\tilde{P}\left(\text{Ebox@Erg}+65=1\right) = (1-p)^3 \times (1-q)^3$$

$$\tilde{P}\left(\text{Ebox@Erg}+65=2\right) = \sum_{n=1}^{3} C(3,n) \times p^n \times (1-p)^{(3-n)} \times (1-q)^3 \tag{1}$$

$$+ \sum_{n=1}^{3} C(3,n) \times q^n \times (1-q)^{(3-n)} \times (1-p)^3$$
$$+ \sum_{n=1}^{2} \sum_{m=1}^{3-n} C(3,n) \times p^n \times (1-p)^{(3-n)} \times C\left((3-n),m\right) \times q^m \times (1-q)^{(3-m)} \tag{2}$$

where $p$ and $q$ represent the expression levels of TAL1 and LYL1, respectively.

However, to remove the bias introduced by simply taking the expression level of a TF as its probability of binding to a motif, we further normalized the resulting probabilities as below:

$$\tilde{Z} = \tilde{P}(\text{Ebox@Erg}+65=1) + \tilde{P}(\text{Ebox@Erg}+65=2) \tag{3}$$

$$P\left(\text{Ebox@Erg}+65=1\right) = \tilde{P}(\text{Ebox@Erg}+65=1)/\tilde{Z} \tag{4}$$

$$P\left(\text{Ebox@Erg}+65=2\right) = \tilde{P}(\text{Ebox@Erg}+65=2)/\tilde{Z} \tag{5}$$

It should be mentioned that the number of the same motifs in a regulatory region was directly taken into account in the estimation of probabilities. One may raise the question of whether this number has such strong power. Specifically, should the exponents in *equations (1)* and *(2)* change

linearly, or less than linearly, along with the increase in the number of Ebox motifs? To address this issue, we replaced all exponents with their square roots and re-run the whole set of simulations (data not shown). Results showed that using the square roots instead of the original numbers (i) caused a more evenly distributed expression of the nine TFs over the hypothetical interval [0, 1], (ii) captured the same trend in gene expression changes in some perturbations (e.g. the AML-ETO simulation), but (iii) led to decreased expression levels of certain TFs in other perturbations (e.g. PU.1 knockdown and GFI1B overexpression), which therefore disagrees with the experimental data. In order to capture a better agreement of computational and experimental results, we directly used the number of motifs to estimate the discrete probability distributions.

## b) Estimating the activity of a regulatory region

The regression coefficient of a regulatory region on a motif family was estimated by normalizing the logarithmic deviation of luciferase activity, e.g. comparing the change of luciferase activity between the wild-type and mutated constructs. For example, when the luciferase activity for the wild-type *Erg* +65 region was set to 100%, the simultaneous mutation of all Ebox or Gfi motifs at this region resulted in increased luciferase activity (181.2% or 475.9%, respectively) (*Figure 3b*). In contrast, simultaneous mutation of all Ets or Gata motifs at this region led to reduced luciferase activity (1.3% or 14.5%, respectively). Based on this information, we estimated the regression coefficient of the *Erg* +65 region on a relevant motif family in the following way:

$$\alpha_i = \log\left(\frac{100}{l_k}\right) \times \left(\sum_k |\log\left(\frac{100}{l_k}\right)|^{-1}\right) \tag{6}$$

where $k \in \{1, ..., 4\}$, $l_1 = 181.2$, $l_2 = 475.9$, $l_3 = 1.3$, $l_4 = 14.5$; accordingly, $\alpha_1 = -0.070$, $\alpha_2 = -0.185$, $\alpha_3 = 0.515$, $\alpha_4 = 0.230$. We can then formulate a linear regression equation as below:

$$\tilde{y} = \alpha_1 x_1 + \alpha_2 x_2 + \alpha_3 x_3 + \alpha_4 x_4 \tag{7}$$

where $\tilde{y}$ denote the estimated luciferase activity of *Erg*+65, and $x_1$, $x_2$, $x_3$ and $x_4$ represent the binding status of Ebox, Gfi, Ets and Gata motifs at *Erg*+65.

However, the minimum and maximum $\tilde{y}$ obtained by the above formula are 0.235 (when $x_1 = x_2 = 2$ and $x_3 = x_4 = 1$) and 1.235 (when $x_1 = x_2 = 1$ and $x_3 = x_4 = 2$). To make the values of $\tilde{y}$ fall in the desired closed interval [0, 1], an intercept of -0.235 has to be introduced into the linear regression model. In addition, a disturbance term has been included in the model to satisfy the generic assumption of conditional linear Gaussian distribution. Finally, the fully defined linear regression model regarding *Erg*+65 is given as:

$$\tilde{y} = c + \alpha_1 x_1 + \alpha_2 x_2 + \alpha_3 x_3 + \alpha_4 x_4 + \varepsilon \tag{8}$$

where $c = -0.235$, $\varepsilon \sim N(0, \sigma^2)$, and $\sigma$ should be a very small value.

## c) Estimating the expression level of a gene

For each gene studied, the regression coefficient of its expression level on a relevant regulatory region was estimated by normalizing the logarithmic deviation of luciferase activity, where deviation refers to the change of luciferase activity compared to an empty vector control.

For example, when setting the luciferase activity of the wild-type constructs to 100%, the luciferase activity of the empty vector controls relative to *Erg*+65, *Erg*+75 and *Erg*+85 wild-types are 1.9%, 1.0% and 15.2%, respectively (*Figure 3b*, *Figure 3—figure supplement 1* and *2*). Based on these data, we estimated the expression level of *Erg* on a relevant regulatory region in the following way:

$$\beta_i = \log\left(\frac{100}{l_k}\right) \times \left(\sum_k |\log\left(\frac{100}{l_k}\right)|\right)^{-1} \tag{9}$$

where $k \in \{1, 2, 3\}$, $l_1 = 1.9$, $l_2 = 1.0$, $l_3 = 15.2$; accordingly, $\beta_1 = 0.379$, $\beta_2 = 0.441$, $\beta_3 = 0.180$. We can then formulate a linear regression equation as below:

$$\tilde{z} = \beta_1 \tilde{y}_1 + \beta_2 \tilde{y}_2 + \beta_3 \tilde{y}_3 \tag{10}$$

where $\tilde{z}$ denote the estimated expression level of *Erg*; and $\tilde{y}_1$, $\tilde{y}_2$ and $\tilde{y}_3$ represent the estimated activities of *Erg*+65, *Erg*+75 and *Erg*+85. Again, a disturbance term has been introduced to the model in order to meet the generic assumption of conditional linear Gaussian distribution. Thus, the fully defined linear regression model regarding *Erg* is given as:

$$\tilde{z} = \beta_1 \tilde{y}_1 + \beta_2 \tilde{y}_2 + \beta_3 \tilde{y}_3 + \varepsilon \qquad (11)$$

where $\varepsilon \sim N(0, \sigma^2)$ and $\sigma$ should be a very small value.

## Statistics

Significance for the results of the luciferase reporter assays was calculated by combining the p-values of each experiment (generated by using the t-test function in Excel) using the Fisher's method, followed by the calculation of Stouffer's z trend if necessary. Significance tests for changes in TF expression levels caused by TF perturbations (both computational and experimental) were evaluated by Wilcoxon rank-sum tests.

## Acknowledgements

We thank the CIMR Flow Cytometry Core facility, especially Dr Chiara Cossetti, for their expertise with cell sorting, Dr Marina Evangelou for her help with statistical analysis of the luciferase assay data, Lucas Greder for advice on cell transfection and stimulating discussions and Cyagen Biosciences and the MRC MHU Transgenic Core for generating transgenic embryos. Thanks are also extended to past and present members of the Göttgens and de Bruijn lab for practical assistance in the analysis of transient transgenic embryos, and to Yoram Groner and Ditsa Levanon for insightful discussions. We are grateful to Barbara L Kee (University of Chicago, USA) for providing the MigRI-Gfi1b vector and Peter Laslo (University of Leeds, UK) for the shPU.1 construct.

## Additional information

### Funding

| Funder | Grant reference number | Author |
|---|---|---|
| Wellcome Trust | 100140 | Judith Schütte<br>Huange Wang<br>Nicola K Wilson<br>Fernando J Calero-Nieto<br>Victoria Moignard<br>Silvia Basilico<br>Sarah J Kinston<br>Rebecca L Hannah<br>Berthold Göttgens |
| Wellcome Trust | G097922 | Judith Schütte<br>Huange Wang<br>Nicola K Wilson<br>Fernando J Calero-Nieto<br>Victoria Moignard<br>Silvia Basilico<br>Sarah J Kinston<br>Rebecca L Hannah<br>Berthold Göttgens |
| Wellcome Trust | G0900729/1 | Judith Schütte<br>Huange Wang<br>Nicola K Wilson<br>Fernando J Calero-Nieto<br>Victoria Moignard<br>Silvia Basilico<br>Sarah J Kinston<br>Rebecca L Hannah<br>Berthold Göttgens |
| Cancer Research UK | C1163/A12765 | Rebecca L Hannah<br>Berthold Göttgens |

| Leukaemia and Lymphoma Research | 12029 | Judith Schütte<br>Nicola K Wilson<br>Fernando J Calero-Nieto<br>Berthold Göttgens |
| --- | --- | --- |
| Biotechnology and Biological Sciences Research Council | BB/I00050X/1 | Nicola K Wilson<br>Berthold Göttgens |
| National Institute for Health Research | | Rebecca L Hannah<br>Berthold Göttgens |
| Medical Research Council | G0900951 | Nicola K Wilson<br>Victoria Moignard<br>Berthold Göttgens |
| MRC Molecular Haematology Unit (Oxford) core award | | Stella Antoniou<br>Andrew Jarratt<br>Joey Riepsaame<br>Mun Chiang Chan<br>Marella FTR de Bruijn |
| Weizmann-UK | | Marella FTR de Bruijn |

The funders had no role in study design, data collection and interpretation, or the decision to submit the work for publication.

### Author contributions

JS, Acquisition of data, Analysis and interpretation of data, Drafting or revising the article; HW, Analysis and interpretation of data, Drafting or revising the article; SA, AJ, JR, FJCN, VM, SB, RLH, Acquisition of data, Analysis and interpretation of data; NKW, Acquisition of data, Drafting or revising the article; SJK, MCC, STN, Acquisition of data; WHO, Conception and design; NB, Conception and design, Analysis and interpretation of data; MFTRdB, Conception and design, Acquisition of data, Analysis and interpretation of data, Drafting or revising the article; BG, Conception and design, Drafting or revising the article

### Author ORCIDs

Berthold Göttgens, http://orcid.org/0000-0001-6302-5705

### Ethics

Animal experimentation: All mice were housed in microisolator cages and provided continuously with sterile food, water, and bedding. All mice were kept in specified pathogen-free conditions, and all procedures were performed according to the United Kingdom Home Office regulations under project licence 70/8406

# Additional files

### Major datasets

The following dataset was generated:

| Author(s) | Year | Dataset title | Dataset URL | Database, license, and accessibility information |
| --- | --- | --- | --- | --- |
| Schütte J, Wang H, Antoniou S, Jarratt A, Wilson NK, Riepsaame J, Calero-Nieto FJ, Moignard V, Basilico S, Kinston SJ, Hannah RL, Chan MC, Nürnberg ST, Ouwehand WH, Bonzanni N, de Bruijn MFTR, Göttgens B | 2016 | A fully validated blood stem/progenitor cell regulatory network reveals mechanisms of cell state stabilisation | http://www.ncbi.nlm.nih.gov/geo/query/acc.cgi?acc=GSE69776 | Publicly available at the NCBI Gene Expression Omnibus (Accession no: GSE69776). |

The following previously published datasets were used:

| Author(s) | Year | Dataset title | Dataset URL | Database, license, and accessibility information |
|---|---|---|---|---|
| Wilson NK, Foster SD, Wang X, Knezevic K, Schütte J, Kaimakis P, Chilarska PM, Kinston S, Ouwehand WH, Dzierzak E, Pimanda JE, de Bruijn MF, Göttgens B | 2010 | Combinatorial transcriptional control in blood stem/progenitor cells: genome-wide analysis of ten major transcriptional regulators | http://www.ncbi.nlm.nih.gov/geo/query/acc.cgi?acc=GSE22178 | Publicly available at the NCBI Gene Expression Omnibus (Accession no: GSE22178). |
| Kassouf MT, Hughes JR, Taylor S, McGowan SJ, Soneji S, Green AL, Vyas P, Porcher C | 2010 | Genome-wide identification of TAL1's functional targets: insights into its mechanisms of action in primary erythroid cells | http://www.ncbi.nlm.nih.gov/geo/query/acc.cgi?acc=GSE18720 | Publicly available at the NCBI Gene Expression Omnibus (Accession no: GSE18720). |
| Heinz S, Benner C, Spann N, Bertolino E, Lin YC, Laslo P, Cheng JX, Murre C, Singh H, Glass CK | 2010 | Simple combinations of lineage-determining transcription factors prime cis-regulatory elements required for macrophage and B cell identities | http://www.ncbi.nlm.nih.gov/geo/query/acc.cgi?acc=GSE21512 | Publicly available at the NCBI Gene Expression Omnibus (Accession no: GSE21512). |
| Wontakal SN, Guo X, Will B, Shi M, Raha D, Mahajan MC, Weissman S, Snyder M, Steidl U, Zheng D, Skoultchi AI | 2011 | A large gene network in immature erythroid cells is controlled by the myeloid and B cell transcriptional regulator PU.1 | http://www.ncbi.nlm.nih.gov/geo/query/acc.cgi?acc=GSE21953 | Publicly available at the NCBI Gene Expression Omnibus (Accession no: GSE21953). |
| Trompouki E, Bowman TV, Lawton LN, Fan ZP, Wu DC, DiBiase A, Martin CS, Cech JN, Sessa AK, Leblanc JL, Li P, Durand EM, Mosimann C, Heffner GC, Daley GQ, Paulson RF, Young RA, Zon LI | 2011 | Lineage regulators direct BMP and Wnt pathways to cell-specific programs during differentiation and regeneration | http://www.ncbi.nlm.nih.gov/geo/query/acc.cgi?acc=GSE29193 | Publicly available at the NCBI Gene Expression Omnibus (Accession no: GSE29193). |
| Tanaka Y, Joshi A, Wilson NK, Kinston S, Nishikawa S, Göttgens B | 2012 | The transcriptional programme controlled by Runx1 during early embryonic blood development | http://www.ncbi.nlm.nih.gov/geo/query/acc.cgi?acc=GSE29515 | Publicly available at the NCBI Gene Expression Omnibus (Accession no: GSE29515). |
| Wu W, Cheng Y, Keller CA, Ernst J, Kumar SA, Mishra T, Morrissey C, Dorman CM, Chen KB, Drautz D, Giardine B, Shibata Y, Song L, Pimkin M, Crawford GE, Furey TS, Kellis M, Miller W, Taylor J, Schuster SC, Zhang Y, Chiaromonte F, Blobel GA, Weiss MJ, Hardison RC | 2011 | Dynamics of the epigenetic landscape during erythroid differentiation after GATA1 restoration | http://www.ncbi.nlm.nih.gov/geo/query/acc.cgi?acc=GSE30142 | Publicly available at the NCBI Gene Expression Omnibus (Accession no: GSE30142). |
| Wu JQ, Seay M, Schulz VP, Hariharan M, Tuck D, Lian J, Du J, Shi M, Ye Z, Gerstein M, Snyder MP, Weissman S | 2012 | Tcf7 is an important regulator of the switch of self-renewal and differentiation in a multipotential hematopoietic cell line | http://www.ncbi.nlm.nih.gov/geo/query/acc.cgi?acc=GSE31221 | Publicly available at the NCBI Gene Expression Omnibus (Accession no: GSE31221). |

| | | | | |
|---|---|---|---|---|
| Zhang JA, Mortazavi A, Williams BA, Wold BJ, Rothenberg EV | 2012 | Dynamic transformations of genome-wide epigenetic marking and transcriptional control establish T cell identity | http://www.ncbi.nlm.nih.gov/geo/query/acc.cgi?acc=GSE31235 | Publicly available at the NCBI Gene Expression Omnibus (Accession no: GSE31235). |
| Doré LC, Chlon TM, Brown CD, White KP, Crispino JD | 2012 | Chromatin occupancy analysis reveals genome-wide GATA factor switching during hematopoiesi | http://www.ncbi.nlm.nih.gov/geo/query/acc.cgi?acc=GSE31331 | Publicly available at the NCBI Gene Expression Omnibus (Accession no: GSE31331). |
| Yu M, Mazor T, Huang H, Huang HT, Kathrein KL, Woo AJ, Chouinard CR, Labadorf A, Akie TE, Moran TB, Xie H, Zacharek S, Taniuchi I, Roeder RG, Kim CF, Zon LI, Fraenkel E | 2012 | Direct recruitment of polycomb repressive complex 1 to chromatin by core binding transcription factors | http://www.ncbi.nlm.nih.gov/geo/query/acc.cgi?acc=GSE33653 | Publicly available at the NCBI Gene Expression Omnibus (Accession no: GSE33653). |
| Ostuni R, Piccolo V, Barozzi I, Polletti S, Termanini A, Bonifacio S, Curina A, Prosperini E, Ghisletti S, Natoli G | 2013 | Latent enhancers activated by stimulation in differentiated cells | http://www.ncbi.nlm.nih.gov/geo/query/acc.cgi?acc=GSE38377 | Publicly available at the NCBI Gene Expression Omnibus (Accession no: GSE38377). |
| Lichtinger M, Ingram R, Hannah R, Müller D, Clarke D, Assi SA, Lie-A-Ling M, Noailles L, Vijayabaskar MS, Wu M, Tenen DG, Westhead DR, Kouskoff V, Lacaud G, Göttgens B, Bonifer C | 2012 | RUNX1 reshapes the epigenetic landscape at the onset of haematopoiesis | http://www.ncbi.nlm.nih.gov/geo/query/acc.cgi?acc=GSE40235 | Publicly available at the NCBI Gene Expression Omnibus (Accession no: GSE40235). |
| Pencovich N, Jaschek R, Dicken J, Amit A, Lotem J, Tanay A, Groner Y | 2013 | Cell-autonomous function of Runx1 transcriptionally regulates mouse megakaryocytic maturation | http://www.ncbi.nlm.nih.gov/geo/query/acc.cgi?acc=GSE45372 | Publicly available at the NCBI Gene Expression Omnibus (Accession no: GSE45372). |
| Org T, Duan D, Ferrari R, Montel-Hagen A, Van Handel B, Kerényi MA, Sasidharan R, Rubbi L, Fujiwara Y, Pellegrini M, Orkin SH, Kurdistani SK, Mikkola HK | 2015 | Scl binds to primed enhancers in mesoderm to regulate hematopoietic and cardiac fate divergence | http://www.ncbi.nlm.nih.gov/geo/query/acc.cgi?acc=GSE47085 | Publicly available at the NCBI Gene Expression Omnibus (Accession no: GSE47085). |
| Calero-Nieto FJ1, Ng FS, Wilson NK, Hannah R, Moignard V, Leal-Cervantes AI, Jimenez-Madrid I, Diamanti E, Wernisch L, Göttgens B | 2014 | Key regulators control distinct transcriptional programmes in blood progenitor and mast cells | http://www.ncbi.nlm.nih.gov/geo/query/acc.cgi?acc=GSE48086 | Publicly available at the NCBI Gene Expression Omnibus (Accession no: GSE48086). |
| Vander Lugt B, Khan AA, Hackney JA, Agrawal S, Lesch J, Zhou M, Lee WP, Park S, Xu M, DeVoss J, Spooner CJ, Chalouni C, Delamarre L, Mellman I, Singh H | 2014 | Transcriptional programming of dendritic cells for enhanced MHC class II antigen presentation | http://www.ncbi.nlm.nih.gov/geo/query/acc.cgi?acc=GSE52773 | Publicly available at the NCBI Gene Expression Omnibus (Accession no: GSE52773). |

| Smith AM, Calero-Nieto FJ, Schütte J, Kinston S, Timms RT, Wilson NK, Hannah RL, Landry JR, Göttgens B | 2012 | Integration of Elf-4 into stem/progenitor and erythroid regulatory networks through locus-wide chromatin studies coupled with in vivo functional validation | http://hscl.cimr.cam.ac.uk/genomic_supplementary.html | Bed and BigWig files can be found under Smith A. et al (2011) |
| --- | --- | --- | --- | --- |
| Wilson NK, Schoenfelder S, Hannah RL, Sánchez Castillo M, Schütte J, Ladopoulos V, Mitchelmore J, Goode DK, Calero-Nieto FJ, Moignard V, Wilkinson AC, Jimenez-Madrid I, Kinston SJ, Spivakov M, Fraser P, Göttgens B | 2016 | Integrated genome-scale analysis of the transcriptional regulatory landscape in a blood stem / progenitor cell model | https://www.ebi.ac.uk/arrayexpress/experiments/E-MTAB-3954/ | Will be publicly available at Array Express once related article published (accession no. E-MTAB-3954). |

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
