## [Decision Letter]

Thank you for submitting your work entitled "A fully validated blood stem/progenitor cell regulatory network reveals mechanisms of cell state stabilisation" for consideration by *eLife*. Your article has been favourably evaluated by Fiona Watt (Senior Editor) and three peer reviewers, one of whom, Amy Wagers, is a member of our Board of Reviewing Editors.

The reviewers have discussed the reviews with one another and the Reviewing editor has drafted this decision to help you prepare a revised submission.

Summary:

The manuscript by Göttgens and colleagues develops an experimentally informed computational strategy to elucidate gene regulatory networks in cells and applies this pipeline to the analysis of key transcription factors in hematopoietic progenitors. The authors identify many new regulatory elements likely to contribute to transcription factor (TF) gene regulation in progenitor cells, and use these to construct a dynamic gene network model. They also test multiple functional effects by knockdown experiments and expression of the oncogenic AML1-ETO dominant negative transcription factor. The manuscript addresses a crucial question – how much information is sufficient to build mammalian gene regulatory network models that capture real biology and have quantitative predictive power? –, and the criteria used to identify cis-regulatory elements of potential interest, as well as the way the authors model regulatory gene interactions in a sophisticated, dynamic Bayesian model that incorporates cis-regulatory site number, TF levels, and known regions of TF occupancy represent significant, strengths of the work. While individual modeling choices can be debated, the authors' approach will certainly be interesting to many readers. They also are very straightforward about assumptions that go into the model that could be subject to revision in future improvements, such as the need to approximate TF levels by the levels of TF coding RNAs, and the baseline assumptions that binding events are independent and purely driven by TF concentrations. Nonetheless, there are some points that the authors still need to address, as detailed below.

Essential revisions:

1) Title/Abstract: The authors claim to provide a "fully validated blood stem/progenitor cell regulatory network", a rather bold statement that is not accurate because: 1. The authors show that some model predictions are incorrect, 2. They do not validate all aspects of the model, and 3. The model includes only 9 transcription factors, and is specific to a single cultured cell line. A more fitting title would be "An integrated gene regulatory network of 9 hematopoietic transcription factors with experimentally validated binding sites".

2) Similar to the above, there are several claims made by the authors that do not (or not entirely) match up with the results presented:

At various points, the authors refer to the "stability of the network" and how this captures the robustness of the biological system; however it is not clear that this is proven by the model and the claims should generally be toned down (including the title).

"Stability of expression levels over time" (paragraph two, subheading “Dynamic Bayesian network modelling can incorporate complex regulatory information and shows stabilization of the HSPC expression state”; Figure 4): Other than PU.1 and perhaps *Erg*, I don't see any TF "rapidly adopting characteristic stable expression levels". All other TFs seem to oscillate irregularly around a baseline more or less equal to the initialization value. Oscillations are a common pattern in real-world systems and (probably) an important component of regulatory mechanisms, but I'm not sure I see any stabilization or these oscillations over the course of the model run.

Figure 6, text: The authors state that there is no expression change observed upon *Gfi1b* overexpression in the computational model. It appears to me that *Scl1* expression is predicted to be affected by the model (losing the highly expressing population), but no change is observed in reality. If the authors claim the change in *Scl1* expression in the model is marginal, some statistics would help to make their point.

Figure 6, text: The observations stated in the text do not match the results apparent from the plots. In the experimental data, *Lyl1* expression is downregulated and there is a subset of cells that completely lose *Lyl1* expression, but there is no change in the computational model. *Scl* expression is erratic in the model – but that is generally the case in all model runs. *Gfi1b* expression is predicted to be affected slightly, but the model fails to capture the fact that a large proportion of cells completely lose expression.

The previous two points highlight the importance of objective (quantitative) comparisons of the model predictions with experimentally derived data. Of course, I don't expect the model predictions to be accurate in all cases (and the authors acknowledge in the Discussion such caveats), but the results should be accurately represented in the text.

3) The reviewers were confused about the numbers of regulatory elements chosen: From 35 candidate regions, 8 are disqualified due to lack of activity, and 2 because they overlap repeats (why do these need to be removed?). This leaves 25 regions, only 9 of which show *LacZ* expression in the respective assays (paragraph two, subheading “in vivo validation of cis-elements as regulatory network nodes connecting 9 HSPC TFs”). In the next part, the text then refers to 23 regions (subheading “ChIP-Seq maps for a second progenitor cell line validate core regulatory interactions”). Figure 1 and Figure 2 only list 23 regions (in black).

4) It appears to be assumed that every mutation within a binding site motif abrogates binding by the respective TF, hence allowing one to reason whether or not a site is functional at all by observing whether the mutation causes a change in target gene activity (subheading “Comprehensive TFBS mutagenesis reveals enhancer-dependent effects of TF binding”). Can it really be assumed that each mutation completely abrogates binding? Moreover, in the text the authors state that for each regulatory element there may be multiple occurrences of any given motif (e.g. six Ets binding sites for *Erg+65*, subheading “Comprehensive TFBS mutagenesis reveals enhancer-dependent effects of TF binding”) – it is not clear how these have been summarized into a single number in Figure 3. The reviewers assume it is the mean response, and Figure 3 gives an idea of the variation of the response to mutations at one specific element, but they wonder how homogeneous the response is elsewhere. Are there cases in which mutations of various sites of the same motif group result in an opposite response? This is not clear from the bar charts, and – if the case – would further complicate the model generated later on.

4) The authors present model prediction results for the knockdown of *Lyl1, Scl*, of both together, as well as the overexpression of *Gfi1b* and the knockdown of PU.1. The latter two are further compared to experimental data (and supposedly supported by, although there are concerns, see previous points). The paper would benefit from the inclusion of all other model predictions for the knockdown/overexpression of any single or any combination of two factors. Such simulations should be straightforward to run with the model at hand and might help to substantiate the claims made by the authors about the system being generally resistant to single-TF perturbations. As it stands we are presented with several examples in favor of the authors' claims, but we do not know whether the model might also make contrary predictions for other factors. We wouldn't expect each case to be supported by experimental evidence (after all, that's the point of having model), but if there are any unexpected observations made in the model runs these need to be explained/harmonized with the current hypothesis.

5) The biggest problem is that in Figure 6, the effects seen with strong perturbations of important TFs are so limited, especially in the experimental data. In part one might imagine that the knockdown is incomplete at the protein level, but in general the authors should more strongly rebut a possible objection that the cross-regulation linkages proposed in the network are not really very significant at all. This is a crucial point, because it affects the whole evaluation of the work that has gone into the model. Also, given the weak effects the authors show when they do actual perturbation experiments, it is important to include at least one additional test – for example knockdown/knockout of *Lyl1*/scl1. It seems that for some of the phenotypes, the authors may be investing marginal results with too much significance, and they are also not clearly articulating or defending their threshold for what is a significant effect. This would be improved by the addition of one more comparison of the model to experimental data.

---

## [Author Response]

*1) Title/Abstract: The authors claim to provide a "fully validated blood stem/progenitor cell regulatory network", a rather bold statement that is not accurate because: 1. The authors show that some model predictions are incorrect, 2. They do not validate all aspects of the model, and 3. The model includes only 9 transcription factors, and is specific to a single cultured cell line. A more fitting title would be "An integrated gene regulatory network of 9 hematopoietic transcription factors with experimentally validated binding sites".*

We very much appreciate the thoughtful comments from the reviewers. We like the suggestion and we wanted to change the title in line with the reviewers suggestion to "An integrated regulatory network of 9 haematopoietic transcription factors with experimentally validated binding sites reveals mechanisms of cell state stabilisation”, but the character limit did not allow us to use this title. We have therefore changed the title to “An experimentally validated network of 9 haematopoietic transcription factors reveals mechanisms of cell state stability”.

*2) Similar to the above, there are several claims made by the authors that do not (or not entirely) match up with the results presented:*

*At various points, the authors refer to the "stability of the network" and how this captures the robustness of the biological system; however it is not clear that this is proven by the model and the claims should generally be toned down (including the title).*

We have modified the text accordingly (subheading “Relative stability to experimental perturbation is recapitulated by the model”) and have updated the title as suggested by the reviewers. Please also see our further responses below referring to the determination of significant expression changes both for the simulated and experimental perturbation experiments.

"Stability of expression levels over time" (paragraph two, subheading “Dynamic Bayesian network modelling can incorporate complex regulatory information and shows stabilization of the HSPC expression state”; Figure 4): Other than PU.1 and perhaps Erg, I don't see any TF "rapidly adopting characteristic stable expression levels". All other TFs seem to oscillate irregularly around a baseline more or less equal to the initialization value. Oscillations are a common pattern in real-world systems and (probably) an important component of regulatory mechanisms, but I'm not sure I see any stabilization or these oscillations over the course of the model run.

We believe this is a misunderstanding which is entirely our fault. Because, we had not shown a key piece of data, which is that if we start the model at different starting levels (for example all TFs at maximum, it still “stabilises” with the same pattern that is shown in the figure. We have now amended the text and introduced a new supplementary figure (see Figure 4—figure supplement 1) to clarify this important point. The reviewer is of course absolutely correct that oscillations are not the same as stable expression. The revised paragraph therefore uses the word “oscillations around a characteristic mean expression level” (paragraph two, subheading “Dynamic Bayesian network modelling can incorporate complex regulatory information and shows stabilization of the HSPC expression state”).

Figure 6, text: The authors state that there is no expression change observed upon Gfi1b overexpression in the computational model. It appears to me that Scl1 expression is predicted to be affected by the model (losing the highly expressing population), but no change is observed in reality. If the authors claim the change in Scl1 expression in the model is marginal, some statistics would help to make their point.

We would like to thank the reviewer for highlighting the fact that a significance calculation was missing. We agree that such calculations are indeed important to clarify which gene expression changes one should focus on. We have therefore now performed statistical tests both on the in silico model and the experimental data contained within the main manuscript (see Figure 5—figure supplement 1). One noteworthy observation from our paper is that, for both the experimental and modelled data, expression profiles for many genes do not follow a normal distribution, but instead show bi- or multimodal expression distributions. To carry out the significance analysis, we therefore performed a Wilcoxon rank-sum test, which measures whether two unpaired, non-normally distributed, samples are likely to come from the same distribution or not.

These significance calculations have revealed a facet of our data that we had previously overlooked, because small fold-changes can be highly significant both in the simulated and experimental perturbations. We have therefore adapted the whole section on perturbations by now introducing the significance calculations early, because even perturbations such as SCL knockout, where there are no large fold-changes, still cause highly significant small-fold alterations to the expression profiles of some genes in our network (paragraph one, subheading “Relative stability to experimental perturbation is recapitulated by the model”). This intriguing observation suggests that our computer model captures aspects of the fine-grained nature of biological networks. It also made us aware that there may be no one perfect way to visualize these small fold-change alterations. The graphs presented in our paper show fitted curves which do not represent simple normal distributions for many of the genes. To provide an alternative visualization, we have now also produced histogram plots (please see Figure 7 for fitted smooth curves and histogram plots). We have now included histogram plots for all perturbations presented in the manuscript in supplementary data (Figure 5—figure supplement 2).

Author response image 1.**DOI:**
http://dx.doi.org/10.7554/eLife.11469.052

To further interrogate the potential relevance of statistically significant small-fold-change alterations in expression, we have generated new experimental data by performing single cell expression analysis following *Scl* knockdown (subheading “Relative stability to experimental perturbation is recapitulated by the model” and revised Figure 5). As expected from our model analysis, the only TF with a significant shift of median expression was *Scl* itself. However, several genes (*Scl, Fli1, Gfi1b*) showed statistically significant small-fold changes, thus showing some overlap in this and the model predictions. Scl-/- mice are not viable because *Scl* is absolutely required for embryonic blood development. However, deletion of *Scl* in adult HSCs only causes minor phenotypes. We believe that it may well be possible that the statistically significant small-fold changes in HSPC network genes may be responsible for the mild phenotypes seen when *Scl* is deleted in adult HSPCs. Moreover, it would have been impossible to detect these changes using conventional expression profiling, because they only become apparent following the statistical analysis of expression distributions generated by assaying lots of single cells. We have adapted the manuscript (subheading “Relative stability to experimental perturbation is recapitulated by the model”) to introduce these new results and analyses, which we believe add substantially to the likely impact of our paper, and are grateful to the reviewers for raising this point.

Figure 6, text: The observations stated in the text do not match the results apparent from the plots. In the experimental data, Lyl1 expression is downregulated and there is a subset of cells that completely lose Lyl1 expression, but there is no change in the computational model. Scl expression is erratic in the model – but that is generally the case in all model runs. Gfi1b expression is predicted to be affected slightly, but the model fails to capture the fact that a large proportion of cells completely lose expression.

We agree with the reviewers that the model does not capture the proportion of cells with complete loss of expression of specific TFs in Figure 6. We believe this may be due to either (i) functionality of the onco-fusion protein not captured by our assumption that its “only” function is as a straightforward dominant-negative one, or (ii) the fact that the computational model is a closed system of only the 9 network TFs, whereas the experimental single cell perturbation is subject to possible knock-on consequences from gene changes outside of the computational network. We have now modified the text accordingly to accurately reflect the experimental data and introduce the caveats mentioned above (subheading “Major perturbations by the A ML-ETO oncoprotein are captured by the network model”).

*The previous two points highlight the importance of objective (quantitative) comparisons of the model predictions with experimentally derived data. Of course, I don't expect the model predictions to be accurate in all cases (and the authors acknowledge in the Discussion such caveats), but the results should be accurately represented in the text.*

We would like to thank the reviewers for their comment. We felt that it was necessary to highlight the caveats of the computational model which is why these were explored in the Discussion of the article as already acknowledged by the reviewers. On reflection, we have now revised the text when describing the results to provide an accurate description of the results. Please see subheadings “Relative stability to experimental perturbation is recapitulated by the model” and “Major perturbations by the A ML-ETO oncoprotein are captured by the network model”.

3) The reviewers were confused about the numbers of regulatory elements chosen: From 35 candidate regions, 8 are disqualified due to lack of activity, and 2 because they overlap repeats (why do these need to be removed?). This leaves 25 regions, only 9 of which show LacZ expression in the respective assays (paragraph two, subheading “in vivo validation of cis-elements as regulatory network nodes connecting 9 HSPC TFs”). In the next part, the text then refers to 23 regions (subheading “ChIP-Seq maps for a second progenitor cell line validate core regulatory interactions”). Figure 1 and Figure 2 only list 23 regions (in black).

We apologise for the lack of clarity regarding the number of regulatory elements which were investigated further. 35 new candidate regions were identified, 8 had previously been shown to not possess in vivo activity within the hematopoietic system. Of the remaining 27 candidates, 2 overlapped with genomic repeat elements (which can cause difficulties during mapping of the ChIP-Seq experiments; this is due to the fact that we can only map reads unambiguously when they have a single unique match in the genome which therefore leads to unreliable mapping across repeat regions). This leaves 25 potential elements; all of these were tested for in vivo activity which resulted in the identification 9 previously unrecognised elements which exhibited *LacZ* expression at hematopoietic sites in E11.5 embryos. These 9 new elements were then combined with the previously published 14 enhancer elements to be investigated further. Of these 23 elements, 3 regions were excluded to retain only those elements which were bound by at least 3 of the 9 TFs and displayed elevated H3K27ac in HPC7 and 416 cells. Overall, 19 cis-regulatory regions were therefore taken forward for further analysis. We have modified the text in subheadings “In vivo validation of cis-elements as regulatory network nodes connecting 9 HSPC TFs” and “ChIP-Seq maps for a second progenitor cell line validate core regulatory interactions” to clarify this issue and included more references for the previously published elements.

*4) It appears to be assumed that every mutation within a binding site motif abrogates binding by the respective TF, hence allowing one to reason whether or not a site is functional at all by observing whether the mutation causes a change in target gene activity (subheading “Comprehensive TFBS mutagenesis reveals enhancer-dependent effects of TF binding”). Can it really be assumed that each mutation completely abrogates binding?*

Our binding site mutations are based on deep biochemical knowledge of the factors involved. All have long been studied in terms of their DNA-binding activity. Both the DNA binding domains and the consensus DNA binding sites have been functionally dissected. Classical transcription factor biology information is available for all these factors. This includes behaviour in electrophoretic mobility shift assays, where competition experiments show that a mutated DNA motif, even when provided in large excess, cannot compete for binding of the factors. We have used this assay ourselves for *Scl, Lyl1, Gata2, Runx1, Fli1, Erg*, Pu.1 (see for example Göttgens et al. EMBO J 2002), and have utilised equivalent data from the published literature for *Meis1* and *Gfi1b*. For all our mutations, we have mutated the key DNA bases involved in DNA-protein interaction, and then scanned the mutated sequence with state-of-the-art computational tools to make sure that no new binding sites were created. We can therefore be confident that conventional TF binding is abrogated in our mutation experiments. We have clarified this issue in the revised text (subheading “Comprehensive TFBS mutagenesis reveals enhancer-dependent effects of TF binding”

Moreover, in the text the authors state that for each regulatory element there may be multiple occurrences of any given motif (e.g. six Ets binding sites for Erg+65, subheading “Comprehensive TFBS mutagenesis reveals enhancer-dependent effects of TF binding”) – it is not clear how these have been summarized into a single number in Figure 3. The reviewers assume it is the mean response, and Figure 3 gives an idea of the variation of the response to mutations at one specific element, but they wonder how homogeneous the response is elsewhere. Are there cases in which mutations of various sites of the same motif group result in an opposite response? This is not clear from the bar charts, and – if the case – would further complicate the model generated later on.

We apologise that the mutation data was not clear. As ChIP-Seq experiments are not able to resolve TF binding at nucleotide level within a given regulatory region and TFs often function in homodimer/trimer complexes which contact multiple motifs, we elected to do the following: When more than one instance of a given motif was present in a single enhancer, we mutated all instances simultaneously. This has now been clarified in the text (subheading “Comprehensive TFBS mutagenesis reveals enhancer-dependent effects of TF binding”).

*4) The authors present model prediction results for the knockdown of Lyl1, Scl, of both together, as well as the overexpression of Gfi1b and the knockdown of PU.1. The latter two are further compared to experimental data (and supposedly supported by, although there are concerns, see previous points).The paper would benefit from the inclusion of all other model predictions for the knockdown / overexpression of any single or any combination of two factors. Such simulations should be straightforward to run with the model at hand and might help to substantiate the claims made by the authors about the system being generally resistant to single-TF perturbations. As it stands we are presented with several examples in favor of the authors' claims, but we do not know whether the model might also make contrary predictions for other factors. We wouldn't expect each case to be supported by experimental evidence (after all, that's the point of having model), but if there are any unexpected observations made in the model runs these need to be explained / harmonized with the current hypothesis.*

We have now performed all permutations of individual TF knockdown/overexpression as well as all possible pairwise combinations of all 9 TFs (a total of 162 simulations, which we have made available in an easy-to-navigate hyperlinked format). This data is now included as Supplementary Data ([Supplementary-material SD6-data]). Importantly, this comprehensive analysis supports our original statement that the network is relatively stable following single TF perturbations. However, as outlined above, the statistical analysis has now given us a more nuanced view of network stability, because it really refers to the large fold-changes, whereas statistically significant small-fold changes also occur following single factor perturbation. We have amended the text to clarify this point (paragraph one, subheading “Relative stability to experimental perturbation is recapitulated by the model”), and also believe that this whole issue is further clarified through inclusion of the new single cell qRT-PCR experiments following *Scl* knock down (Figure 5).

5) The biggest problem is that in Figure 6, the effects seen with strong perturbations of important TFs are so limited, especially in the experimental data. In part one might imagine that the knockdown is incomplete at the protein level, but in general the authors should more strongly rebut a possible objection that the cross-regulation linkages proposed in the network are not really very significant at all. This is a crucial point, because it affects the whole evaluation of the work that has gone into the model. Also, given the weak effects the authors show when they do actual perturbation experiments, it is important to include at least one additional test – for example knockdown/knockout of Lyl1/scl1. It seems that for some of the phenotypes, the authors may be investing marginal results with too much significance, and they are also not clearly articulating or defending their threshold for what is a significant effect. This would be improved by the addition of one more comparison of the model to experimental data.

As already outlined above, we have now included a significance calculation (Wilcoxon rank-sum test) for all the perturbations in the main article (both in silico model data and experimental). This has given us a more nuanced view of the data, as it has highlighted that small fold-changes, that we previously discounted, can be highly significant. Significant cross-regulatory linkages indeed occur in both the PU.1 knockdown and *Gfi1b* overexpression (both simulation and experiment), that were part of our original submission. We have therefore adapted the whole section on perturbations by now introducing the significance calculations early, because even the single TF perturbations can cause highly significant small-fold alterations to the expression profiles of other genes in our network. This intriguing observation suggests that our computer model captures aspects of the fine-grained nature of biological networks. We followed the reviewers’ advice and attempted to complete a double knockdown of both *Scl* and *Lyl1*, but unfortunately we could not achieve reliable knock down of both factors in single cells. Nevertheless, we did achieve good knock down of *Scl* which we have now included as an additional result in the manuscript (see page 10 and revised Figure 5). Single cell qRT-PCR analysis of *Scl* knockdown cells showed statistically significant small-fold changes in some of the other TS (*Gfi1b* and *Fli1*). Although there was no perfect match between model prediction and experimental validation, there was some overlap. Scl-/- mice are not viable because *Scl* is absolutely required for embryonic blood development. However, deletion of *Scl* in adult HSCs only causes minor phenotypes. We believe that it may well be possible that the statistically significant small-fold changes in HSPC network genes may be responsible for the mild phenotypes seen when *Scl* is deleted in adult HSPCs (or similarly the mild phenotypes following other single TF deletion in HSCs). Importantly, it would have been impossible to detect these changes using conventional expression profiling, because they only become apparent following the statistical analysis of expression distributions generated by assaying lots of single cells.

As requested by the reviewers, our revised manuscript now (i) includes one more comparison of experimental knockdown with computer simulation data, (ii) provides further insights into cross-regulatory relationships, and (iii) introduce a statistical test to identify significant expression changes. We are grateful to the reviewers for raising these points, because we believe that the additional experiments and analysis have allowed us to strengthen our paper significantly.